# A Foxf1-Wnt-Nr2f1 cascade promotes atrial cardiomyocyte differentiation in zebrafish

Ugo Coppola[1¤], Bitan Saha[1], Jennifer Kenney[1], Joshua S. Waxman[1,2,3]*

**1** Molecular Cardiovascular Biology Division and Heart Institute, Cincinnati Children's Hospital Medical Center, Cincinnati, Ohio, United States of America, **2** Developmental Biology Division, Cincinnati Children's Hospital Medical Center, Cincinnati, Ohio, United States of America, **3** Department of Pediatrics, University of Cincinnati, College of Medicine, Cincinnati, Ohio, United States of America

¤ Current address: Department of Biological Sciences, Florida Gulf Coast University, Ft. Myers, Florida, United States of America

* joshua.waxman@cchmc.org

**Data Availability Statement:** All data supporting this study are provided within the manuscript and its Supporting Information.

**Funding:** Work in the manuscript was supported by National Institutes of Health (NIH) grants R01

## Abstract

Nr2f transcription factors (TFs) are conserved regulators of vertebrate atrial cardiomyocyte (AC) differentiation. However, little is known about the mechanisms directing Nr2f expression in ACs. Here, we identified a conserved enhancer 3' to the *nr2f1a* locus, which we call *3'reg1-nr2f1a* (*3'reg1*), that can promote Nr2f1a expression in ACs. Sequence analysis of the enhancer identified putative Lef/Tcf and Foxf TF binding sites. Mutation of the Lef/Tcf sites within the *3'reg1* reporter, knockdown of Tcf7l1a, and manipulation of canonical Wnt signaling support that Tcf7l1a is derepressed via Wnt signaling to activate the transgenic enhancer and promote AC differentiation. Similarly, mutation of the Foxf binding sites in the *3'reg1* reporter, coupled with gain- and loss-of-function analysis supported that Foxf1 promotes expression of the enhancer and AC differentiation. Functionally, we find that Wnt signaling acts downstream of Foxf1 to promote expression of the *3'reg1* reporter within ACs and, importantly, both Foxf1 and Wnt signaling require Nr2f1a to promote a surplus of differentiated ACs. CRISPR-mediated deletion of the endogenous *3'reg1* abrogates the ability of Foxf1 and Wnt signaling to produce surplus ACs in zebrafish embryos. Together, our data support that downstream members of a conserved regulatory network involving Wnt signaling and Foxf1 function on a *nr2f1a* enhancer to promote AC differentiation in the zebrafish heart.

## Author summary

Vertebrate hearts are comprised of atrial chambers, which receive blood, and ventricular chambers which expel blood, whose functions need to be coordinated for proper blood circulation. During development, different genetic programs direct the development of these chambers within the vertebrate heart. Members of a family of genes called *Nr2fs* are conserved regulators of atrial chamber development in vertebrates, with mutations in *Nr2f2* of humans being associated with congenital heart defects affecting the atrium. Here, we examine how the gene *nr2f1a*, which is required for normal atrial chamber

HL168790 and R01 HL137766 to J.S.W., by American Heart Association (AHA) postdoctoral fellowship 831018 to U.C., and by AHA Summer Undergraduate Research Fellowship (SURF) grant 18UFEL33930019 to J.K. The funders had no role in study design, data collection and analysis, decision to publish, or preparation of the manuscript.

**Competing interests:** The authors have declared that no competing interests exist.

development in the model zebrafish, is regulated. Using tools, including transgenic reporter lines and genetic mutants, we identify that factors previously shown to regulate atrial chamber development in mammals have conserved roles regulating a genetic element that promotes *nr2f1a* expression within developing atrial cells. Since there is a lack of understanding regarding regulation of *Nr2f* genes during vertebrate atrial cell development, our work provides insights into the conservation of genetic networks that promote heart development in vertebrates and if perturbed could underlie congenital heart defects.

## Introduction

Integrated regulatory networks drive chamber-specific atrial and ventricular cardiomyocyte programs from spatially and temporally defined progenitor populations [1–3]. Key factors within these regulatory networks are nuclear receptor subfamily 2 group F members (Nr2fs; formerly called Coup-tfs), which are critical regulators of vertebrate atrial cardiomyocyte (AC) differentiation during earlier stages of cardiogenesis [4] and subsequently maintain AC identity at later stages [4–6]. Although both Nr2f1 and Nr2f2 paralogs in humans and mice are expressed in ACs [5–7], genetic mapping and functional analysis in cell culture and animal models support that Nr2f2 is the predominant Nr2f that is required for AC differentiation in mammals. In humans, genetic mapping studies have associated mutations in *NR2F2* with structural congenital heart defects (CHDs), including atrioventricular septal defects (AVSDs) [7–9], a malformation predominantly caused by failure of ACs at the venous pole to differentiate and form the atrial septum [10]. Consistent with the necessity of Nr2f2 in mammalian AC differentiation, global *Nr2f2* knockout (KO) mice have small, dysmorphic atrial chambers that lack septa [4]. Furthermore, NR2F2 has a greater requirement promoting atrial differentiation compared to NR2F1 in human embryonic and induced pluripotent stem cell-derived models [11]. In contrast to mammalian Nr2fs, zebrafish *nr2f1a* is the functional homolog of mammalian Nr2f2, as it is required for AC differentiation and the maintenance of AC identity [12,13]. Hence, loss of *nr2f1a* results in zebrafish embryos with small atria reminiscent of the atrial defects of Nr2f2 global KO mice [4,11].

While studies in human stem cell-derived cardiomyocytes and mice cumulatively have demonstrated that Nr2f2 is a critical node within a regulatory network that simultaneously promotes atrial differentiation and represses ventricular differentiation within cardiomyocytes [4,5,11,14], we have limited understanding of the upstream regulation of *Nr2f* genes that lead to the proper differentiation of vertebrate ACs, with a variety of somewhat disparate signals and factors showing they can regulate Nr2f expression. In many *in vivo* and *in vitro* developmental contexts, retinoic acid (RA) signaling, which is required to limit heart size and promote differentiation of the posterior second heart field (SHF) [15–18], is necessary and sufficient to promote *Nr2f1* and *Nr2f2* expression [19–22]. Furthermore, RA signaling is sufficient to induce AC differentiation in many *in vitro* induced pluripotent stem cell protocols [23–25]. However, despite candidate RA response elements and epigenetic changes near the promoter [22], it has not been shown if RA signaling directly regulates *Nr2f* expression within differentiating ACs [26]. In human embryonic stem cell-derived cardiomyocytes, the TF Isl1, which is required for SHF differentiation, negatively regulates *NR2F1* downstream of RA signaling [27]. Additionally, Tbx20, a conserved regulator of cardiomyocyte proliferation [28], is able to promote *Nr2f2* in the ACs of mice through an evolutionarily conserved enhancer [28]. However, as *Isl1* and *Tbx20* are expressed more broadly, and not specifically expressed in differentiating AC progenitors at the venous pole or differentiated ACs, how these TFs coordinate with

other signals to promote Nr2f2 expression in ACs is not clear. Similarly, Hedgehog (Hh) signaling has been shown to be able to promote mouse *Nr2f2* via Gli TFs binding to its promoter *in vitro* [29], although this has not been shown in heart development. Taken together, it is interesting that many of these factors that regulate Nr2fs, in particular RA and Hh signaling, participate in a regulatory network that controls the timing of SHF progenitor differentiation at the venous pole [30,31]. Elegant work from murine and stem cell models has demonstrated that Hh signaling is an upstream factor in a regulatory network involving RA and Wnt signaling, and Foxf1/2 and Tbx5 TFs that controls the proper timing of SHF progenitor differentiation at the venous pole of the heart [30,32]. Loss of most of these factors results in AVSDs in murine models and humans [33,34], again consistent with the failure of proper AC differentiation and the impairment of dorsal mesenchymal protrusion development, which contributes to the atrial septum. Despite this understanding of upstream factors in this regulatory network directing AC differentiation at the venous pole in mammals, the mechanisms promoting Nr2f expression in ACs and if there is a connection to the Hh-dependent regulatory network controlling posterior SHF addition are not understood.

Here, we sought to better understand regulatory networks directing Nr2f expression in the differentiation of ACs. Examining regions of open chromatin from Assay for Transposase-Accessible Chromatin (ATAC)-seq analysis of embryonic ACs [35] that were conserved within vertebrates [13], we identified an enhancer 3' to the *nr2f1a* locus, which we named *3'reg1-nr2f1a* (*3'reg1*), that was able to promote expression within ACs of stable transgenic lines. A Lef/Tcf binding site and multiple Foxf binding sites were found within the enhancer. Mutation or deletion of the Left/Tcf site led to ectopic *3'reg1* reporter activation within the heart, while mutation or deletion of the Foxf sites led to decreased *3'reg1* reporter expression within ACs. Complementary gain- and loss-of-function experiments manipulating Wnt signaling and Foxf1 coupled with epistasis analysis support a model whereby Wnt signaling, which functions at the level or downstream of Foxf1, relieves Tcf7l1a-mediated repression of the *3'reg1* enhancer and promote AC differentiation. However, in contrast to the established role of Hh signaling functioning upstream of these factors in venous SHF differentiation in mammals, Hh was found to modestly repress transgenic *3'reg1* reporter expression and did not affect AC differentiation. Consistent with Nr2f1a functioning downstream of Wnt signaling and Foxf1, neither is sufficient to promote a surplus of ACs in zebrafish *nr2f1a* mutants. Furthermore, CRISPR-mediated deletion of the endogenous *3'reg1* enhancer showed that this abrogated the ability of Wnt signaling and Foxf1 to enhance the number of differentiated ACs. Cumulatively, our results support that downstream components of a regulatory network involving Wnt signaling and Foxf1, which control differentiation of the SHF in mammals, promote AC differentiation via directing zebrafish *nr2f1a* expression through a conserved 3' enhancer.

## Results

### A *nr2f1a* enhancer drives expression in ACs

To decipher regulatory networks that control *Nr2f* gene expression with respect to its role in heart development, we first examined ~1.5 kb upstream of the transcription start site (TSS) of *nr2f1a* as well as the 5'-untranslated region (5'-UTR) for the ability to promote expression in ACs (**S1A Fig** and **S1 Table**). Only 1.5 kb upstream were chosen as to not include an adjacent lincRNA, which could complicate analysis. The 5'-UTR was included because we previously identified a conserved RA response element in the *nr2f1a* 5'-UTR [13]. Five different constructs encompassing the 1.5 kb of the *nr2f1a* promoter and different lengths of 5'-UTR were examined with reporter vectors driving GFP (**S1A–S1F Fig**). In stable transgenic lines, all the

cloned regions showed similar broad expression throughout the embryos, including in the nervous system, somites, and the heart (**S1B–S1F Fig**). Additionally, the promoter constructs were not responsive to manipulation of RA signaling. Thus, regulatory elements contained within the proximal promoter do not appear to drive *nr2f1a* expression specifically in ACs, consistent with the broader transcriptional regulation generally found with proximal promoters [36], implying more distal *cis*-regulatory elements (CREs) likely promote AC-specific *nr2f1a* expression in zebrafish.

Given the conservation of their expression within ACs, we next postulated that *Nr2f* genes might have conserved CREs that promote their AC expression. To identify conserved CREs that promote expression specifically in the ACs, we examined regions of open chromatin found in our recently reported ATAC-seq on isolated ACs from 48 hours post-fertilization (hpf) embryos [35] for conservation using genomic alignments of all vertebrate *Nr2f1* loci with VISTA plots (**Figs 1A–1C, S2A, S2B, S2D, S2F, and S2H**). Although there is a conserved gene desert 5' to vertebrate *Nr2f* genes, we did not find conservation of sequences within regions of open chromatin in the ~700kb 5' distal the zebrafish *nr2f1a*. The regions of open chromatin with conserved sequence in vertebrate genomes were primarily within ~20kb 5' and 3' of the *nr2f1a* locus, with the conservation in 3' regions of open chromatin extending to the syntenic gene *fam172a* [37] (**S2A Fig**). We identified 7 conserved open regions flanking *nr2f1a* (**Figs 1A–1C, S2A, S2B, S2D, S2F, S2H and S3A–S3D.**), which were subsequently analyzed via cloning into a GFP reporter vector [38]. Two of the putative enhancers did not drive significant GFP expression in transient transgenic embryos and were not studied further. The rest of the putative enhancers were named according to their 5' and 3' positions relative to *nr2f1a* (**Figs 1A and S2A**). Those that did drive tissue-specific expression in transient transgenics were raised to confirm their expression in stable transgenic lines. Two of the transgenic enhancer reporters showed expression within subdomains that are reminiscent of the endogenous extra-cardiac *nr2f1a* expression [39]. *5'reg1-nr2f1a* was expressed in the anterior hindbrain and branchial arches (**S2C Fig**). *3'reg4-nr2f1a* was expressed in the medial eye, nasal pits, and anterior branchial arch (**S2I Fig**). Two putative enhancers, *3'reg2-nr2f1a* and *3'reg3-nr2f1a*, were expressed within the heart by 48 hpf, but did not show atrial-specific expression (**S2E and S2G Fig**). However, the *3'reg1-nr2f1a* enhancer (henceforth referred to as *3'reg1*) showed expression specifically within ACs by 24 hpf and overlap with endogenous Nr2f1a within the development heart through 72 hpf (**Figs 1D–1H and S4**). Moreover, the strongest expression of the *3'reg1* reporter was at the venous pole of the atrium and it tapered toward the atrioventricular canal/outflow pole of the atrium (**Figs 1E–1H and S4**). Although *nr2f1a* expression is first detectable by early somitogenesis stages in the ALPM [13], with the present analysis we did not identify CREs that drove expression during earlier stages in the ALPM. Thus, we identified a conserved putative *nr2f1a* enhancer that is able to promote expression within zebrafish ACs by the early heart tube stage.

## Wnt signaling promotes Nr2f1a expression in ACs

To determine signals that could regulate *3'reg1*, we performed TF binding site (TFBS) analysis using the CIS-BP [40,41], TomTom [42,43], and JASPAR [44] databases. Among the different putative TFBSs identified within *3'reg1* were a nuclear hormone receptor (NHR) binding site, a Lef/Tcf binding site, and multiple Foxf binding sites (**Figs 1C and S5A**), reflecting signals, including RA and Wnt signaling, that are involved in promoting the proper timing of AC differentiation from the posterior SHF progenitors in mice [15,16,32,45]. Furthermore, there was enriched conservation of nucleotides in the regions of the enhancers harboring the potential binding sites, although there was less conservation over the putative Lef/Tcf binding site

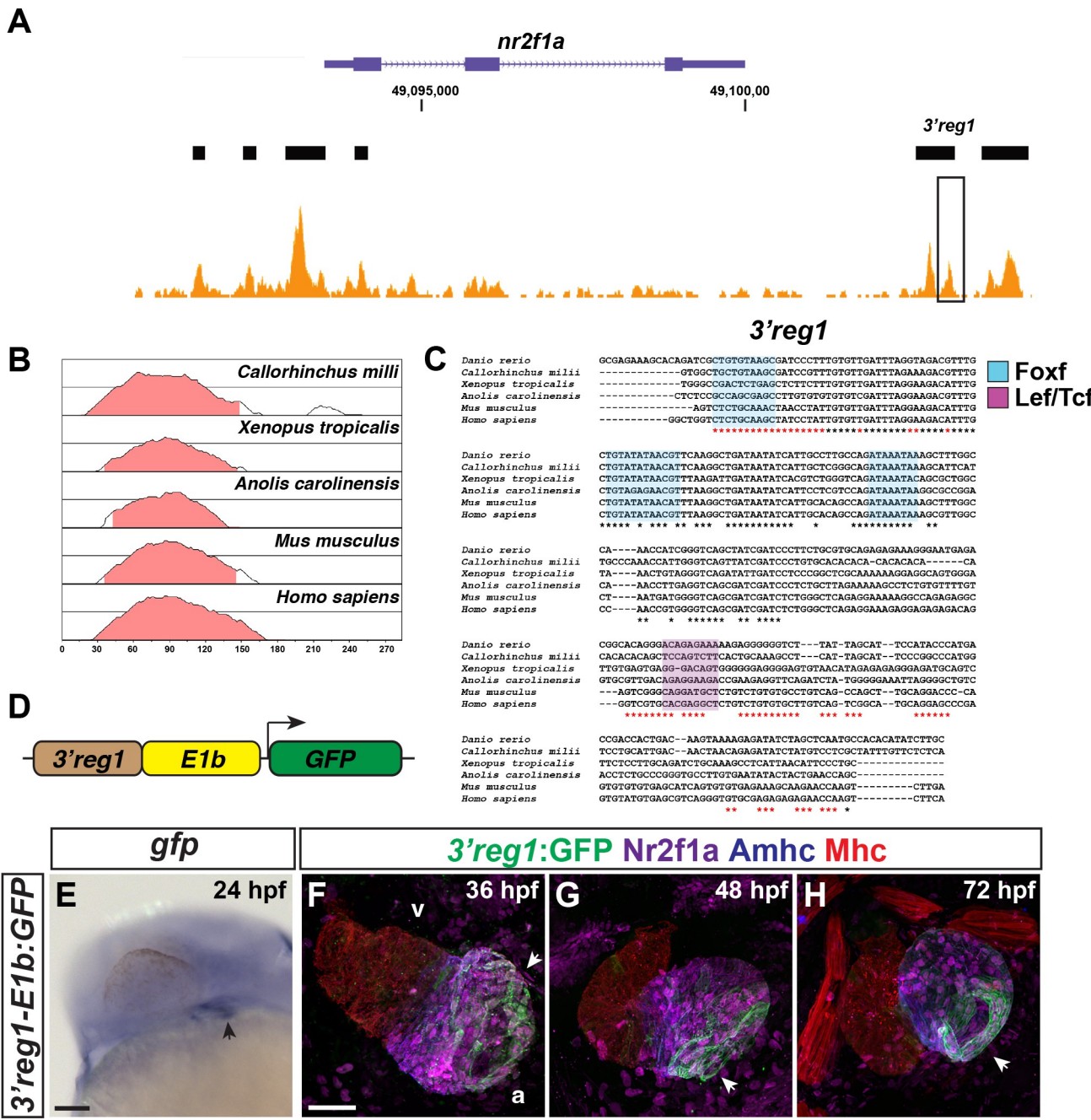

**Fig 1. The *3'reg1-nr2f1a* enhancer is expressed in the atrium. A)** ATAC-seq profiling from ACs showing regions of open chromatin near the *nr2f1a* locus (purple). Black bars indicate called regions of open chromatin. Black box indicates *3'reg1*. **B)** VISTA plot showing conservation of zebrafish *3'reg1* enhancer in *Callorhincus milli* (Australian ghostshark), *Xenopus tropicalis* (Tropical clawed frog), *Anolis carolinensis* (Green Anole), *Mus musculus* (House mouse), and *Homo sapiens* (human). Pink indicates >50% conservation of regulatory regions with zebrafish *3'reg1*. Median lines in individual VISTA plots indicate 75% conservation. **C)** Clustal alignment of *3'reg1* from the VISTA plot in indicated species. Black asterisks indicate conserved nucleotides. Red asterisks indicate partially conserved nucleotides. Conserved putative binding sites for Foxf (blue shade) and Lef/Tcf (purple shade) TFs. **D)** Schematic of the *3'reg1:GFP* reporter vector. **E)** *In situ* hybridization (ISH) for *gfp* in transgenic *3'reg1:GFP* embryo at 24 hpf (n = 24). Venous pole of the heart tube (black arrowhead). View is lateral with anterior left and dorsal up. Scale bar: 200μm. **F-H)** Confocal images of hearts from transgenic *3'reg1:GFP* embryos stained for *3'reg1*:GFP (green), Nr2f1a (magenta), Amhc (ACs–blue), and Mhc (pan-cardiac–red). 36 hpf (n = 4), 48 hpf (n = 4), 72 hpf (n = 4). v–ventricle. a–atrium. *3'reg1:GFP* in the atria of the hearts (white arrowheads). Images are frontal views with the arterial pole up. n indicates the number of embryos examined for representative experiment. Scale bar: 50 μm. **** indicate P < 0.0001.

(**Fig 1C**). We next systematically interrogated the requirement of each of these sites in promoting *3'reg1*:*GFP* expression in ACs. Since RA signaling potentially regulates *Nr2f* expression during atrial differentiation [13,25,46], we initially examined the requirement of the NHR site (**S5A–S5C Fig**). However, deletion of the NHR site did not alter expression of the reporter compared to the unaltered *3'reg1*:*GFP* reporter in transient transgenic embryos nor did modulation of RA signaling via treatment with RA and the RA signaling inhibitor DEAB affect expression in stable transgenic *3'reg1*:*GFP* lines (**S5D and S5E Fig**). Therefore, RA signaling does not appear to regulate *nr2f1a* expression through this enhancer.

Lef/Tcf TFs are mediators of canonical Wnt signaling [47,48], with numerous *in vivo* and *in vitro* studies having shown Wnt signaling controls differentiation of ACs [49,50]. Furthermore, we have shown that increased Wnt signaling during later somitogenesis immediately prior to the formation of the heart tube is necessary and sufficient to specifically produce an increase in ACs in zebrafish embryos [51]. Additionally, studies have demonstrated that Wnt signaling is expressed in a gradient extending from the venous pole of the zebrafish heart [52,53], which is reminiscent of the *3'reg1*:*GFP* reporter. We found that deletion and targeted mutation of the Lef/Tcf site resulted in an expansion of the reporters throughout the hearts, as well as increased intensity of the GFP expression, by 48 hpf relative to the *3'reg1*:*GFP* in transient transgenic embryos (**Fig 2A–2G**). As this result suggests the site may be used to restrict enhancer expression, we reasoned that Tcf7l1a (formerly called Tcf3a), a transcriptional repressor whose repression is relieved by Wnt signaling [54] that we have previously implicated in early cardiomyocyte development [55], may be limiting the expression of the reporter within the heart at these stages of cardiogenesis. To determine if Tcf7l1a limits expression of the reporter, *3'reg1*:*GFP* embryos were injected with an established morpholino (MO) that identically phenocopies the zebrafish *tcf7l1a* mutants [56]. We found that Tcf7l1a depletion produced pan-cardiac expression with increased intensity of the *3'reg1*:*GFP* reporter (**Fig 2H–2K**), suggesting that Tcf7l1a is required to limit the location and levels of *3'reg1* reporter expression within the heart. Given the effects Tcf7l1a on the *3'reg1* reporter, we next examined the number of endogenous Nr2f1a[+] cardiomyocytes within hearts, as one proxy for Nr2f1a expression levels, as well as the localization and the intensity of Nr2f1a[+] in the nuclei in Tcf7l1a depleted embryos. We found that by 48 hpf the hearts of Tcf7l1a depleted embryos had an increased number of Nr2f1a[+]/Amhc[+] atrial cardiomyocytes and the intensity of Nr2f1a expression in nuclei was increased (**Fig 3A–3B'''**). While there was not a dramatic expansion of Nr2f1a[+] cardiomyocytes in the Tcf7l1a-depleted hearts similar to the *3'reg1* reporter, we observed a few low expressing, Nr2f1a[+] nuclei within ventricular cardiomyocytes (VCs) of Tcf7l1a-depleted embryos (**Fig 3B**, **3B'**, **and 3K**). Thus, our data suggest that Tcf7l1a limits expression of the *3'reg1*:*GFP* reporter and the number and levels of Nr2f1a[+] cardiomyocytes within the heart, although the induction of ectopic Nr2f1a is modest relative to the *3'reg1* reporter.

The activation of the transgenic *3'reg1*:*GFP* reporter via Tcf7l1a depletion would predict Wnt signaling should also regulate the *3'reg1*:*GFP* reporter expression within the heart. Additionally, the early depletion of Tcf7l1a, which can affect early patterning [55], might obfuscate the effects on the heart observed at the relatively later stages. Therefore, we manipulated Wnt signaling in *3'reg1*:*GFP* embryos at the 20 somite (s) stage with treatments of BIO and XAV939 (XAV), which respectively activate and inhibit Wnt signaling [57]. The 20s stage was chosen because we previously found that modulation of Wnt signaling could specifically affect AC number slightly prior to this stage at ~16s [51,55]. Although the effect of Wnt signaling on AC differentiation was not examined previously at the 20s stage, this stage was also just prior to when we can detect transgenic *3'reg1*:*GFP* reporter expression. We found that by 48 hpf the hearts of the embryos treated with BIO showed pan-cardiac GFP expression with increased

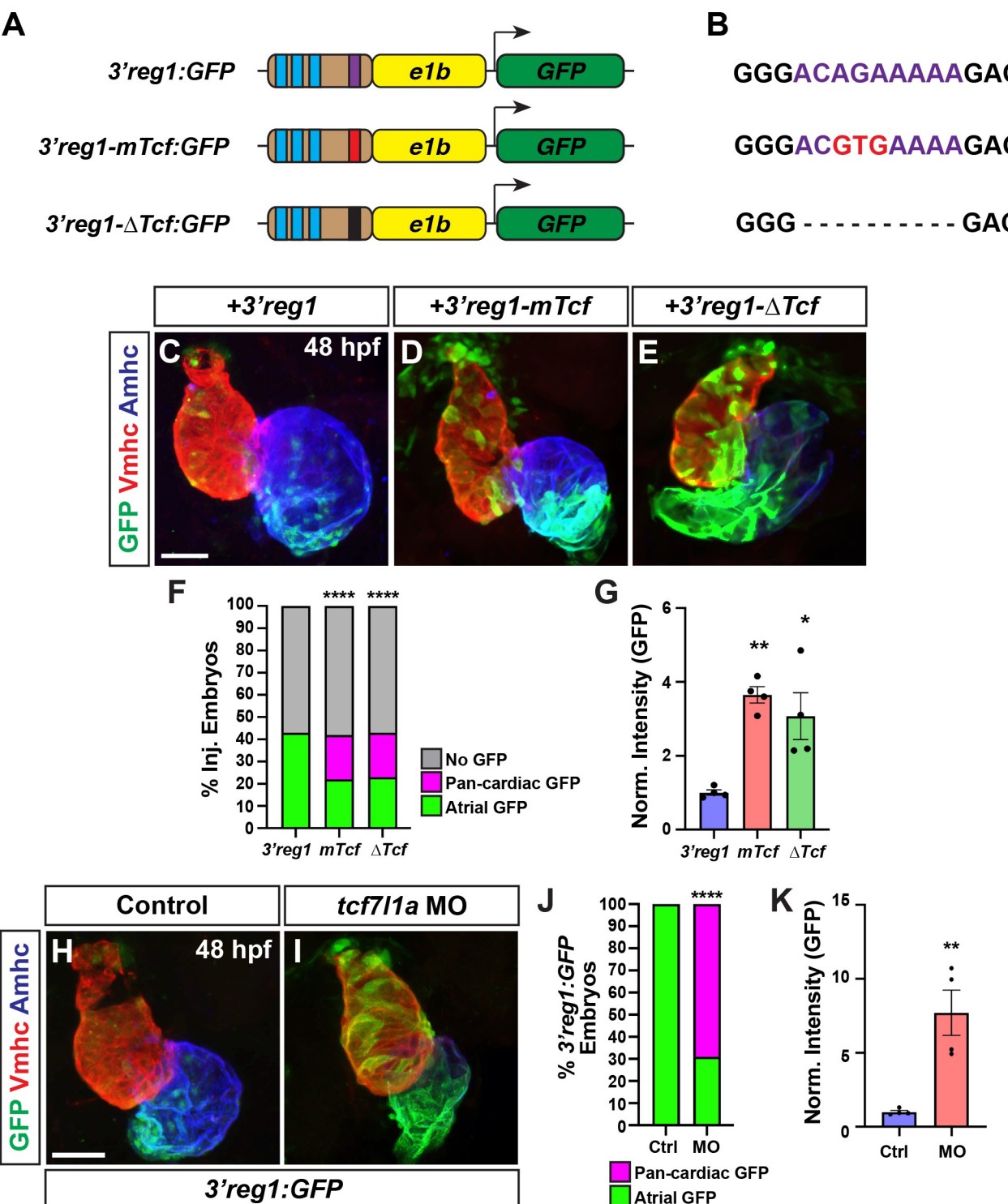

**Fig 2. Tcf7l1a restricts *3'reg1* reporter expression within the heart. A)** Schematics of *3'reg1:GFP* reporter constructs. Foxf sites (blue), Lef/Tcf site (purple), mutated Lef/Tcf site (red), and deleted Lef/Tcf site (black). **B)** Sequences of the WT (purple), mutated (red), and deleted Left/Tcf sites within the *3'reg1* enhancer. **C-E)** Confocal images of hearts from embryos injected with the *3'reg1:GFP*, *3'reg1-mTcf:GFP*, *3'reg1-ΔTcf:GFP* constructs. **F)** The percentage of transient transgenic embryos with reporter atrial, pan-cardiac, and lacking expression in their hearts. *3'reg1:GFP* (n = 200); *3'reg1-mTcf:GFP* (n = 101); *3'reg1-ΔTcf:GFP* (n = 102). **G)** Normalized intensity of GFP expression in hearts from *3'reg1:GFP* (n = 4), *3'reg1-mTcf*: *GFP* (n = 4), and *3'reg1-ΔTcf:GFP* (n = 4) embryos. **H,I)** Confocal images of hearts from control and *tcf7l1a* MO-injected transgenic *3'reg1:GFP* embryos. **J)** The percentage of stable *3'reg1:GFP* embryos with reporter atrial and pan-cardiac expression in their hearts. Control (n = 152); *tcf7l1a-*

*MO* (n = 92). Images are frontal views with the arterial pole up. Hearts are stained for *3'reg1*:GFP (green), Vmhc (red), Amhc (blue). Scale bars: 50 μm. **K)** Normalized intensity of GFP expression in hearts from control and *tcf7l1a* MO-injected stable *3'reg1:GFP* embryos. Control (n = 4), *tcf7l1a* MO (n = 4).* indicate P < 0.02, ** indicate P<0.003, **** indicate P < 0.0001.

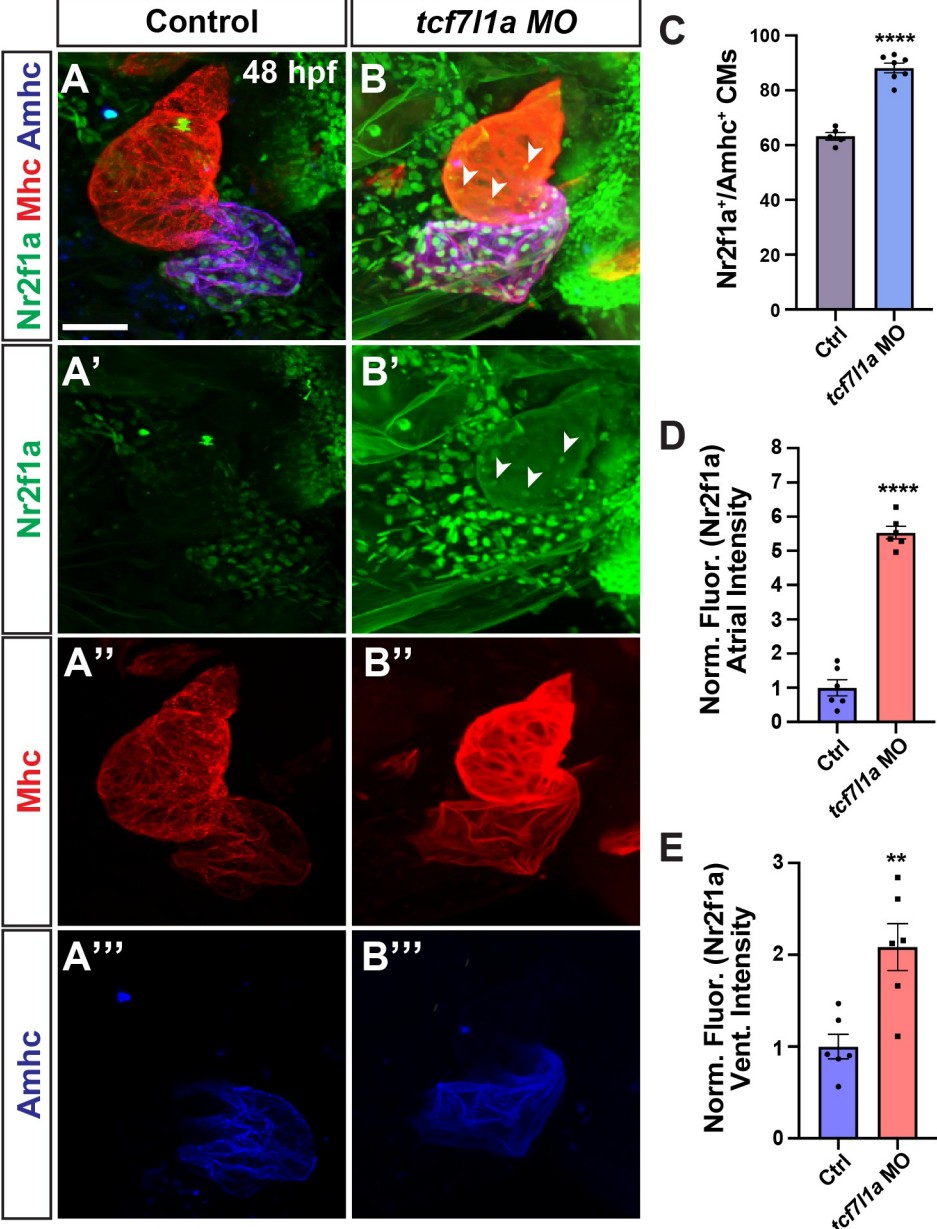

**Fig 3. Tcf7l1a limits Nr2f1a$^+$ cardiomyocytes within the heart. A-B''')** Confocal images of hearts from control and *tcf7l1a* MO-injected embryos. Nr2f1a (green), Mhc (red), Ahmc (blue). Images are frontal views with the arterial pole up. Arrowheads in B and B' indicate Nr2f1a$^+$ nuclei in the ventricle. Scale bar: 50 μm. **C)** The number of Nr2f1a$^+$/Amhc$^+$ cardiomyocytes in the hearts of control and *tcf7l1a* MO-injected embryos. Control (n = 6); *tcf7l1a* MO-injected (n = 7) embryos. **D)** Normalized intensity of Nr2f1a expression in atria of control and *tcf7l1a* MO-injected embryos. Control (n = 6); *tcf7l1a* MO-injected (n = 6) embryos. **E)** Normalized intensity of Nr2f1a expression in ventricles of control and *tcf7l1a* MO-injected embryos. Control (n = 6); *tcf7l1a* MO-injected (n = 6) embryos. ** indicates P = 0.0038; **** indicates P< 0.0001.

intensity (**Fig 4A, 4B, 4D, and 4E**), similar to perturbation of the Lef/Tcf sites in the *3'reg1*: *GFP* reporter (**Fig 2C–2G**) and Tcf7l1a depletion (**Fig 2H–2K**), while XAV abrogated GFP expression within the atria (**Fig 4A and 4C–4E**).

Since our data supported that Wnt signaling also activates the *3'reg1:GFP* reporter, likely via derepression of Tcf7l1a [54], and Tcf7l1a depletion leads to increased Nr2f1a within the atria, we tested if modulation of Wnt signaling at these stages impacts Nr2f1a within the heart. The number and intensity of Nr2f1a[+] cardiomyocyte nuclei within the hearts of 48 hpf embryos treated with BIO and XAV beginning at the 20s stage were quantified. Similar to Tcf7l1 depletion, BIO treatment at the 20s stage produced a surplus of Nr2f1a[+]/Ahmc[+] cardiomyocytes (ACs) and intensity of Nr2f1 nuclear staining by 48 hpf, with some hearts that had a few ectopic Nr2f1a[+] nuclei in VCs (**Figs 4F**, **4G**, **4I**, **S6A**, **S6B**, **S6D**, **and S6E**). Conversely, XAV treatment caused a decrease in Nr2f1a[+]/Amhc[+] cardiomyocytes (ACs) within the atrium, with the Amhc[+] ACs having a reduction in the intensity of Nr2f1a expression in their nuclei (**Figs 4F**, **4H**, **4I**, **S6A**, **S6C**, **and S6D**). To quantitatively examine *nr2f1a* expression in the hearts following manipulation of Wnt signaling, we performed RT-qPCR on cDNA from hearts isolated at 24 hpf following treatment of BIO and XAV. We found that BIO was able to induce *nr2f1a* expression in these hearts, while XAV treatment caused a decrease in its expression (**S7A and S7B Fig**). Based on the effects of Wnt signaling on Nr2f1a in ACs and our previous observations that manipulation of Wnt signaling at similar somitogenesis stages specifically affected AC production [51], we tested the impact of Wnt manipulation on ACs using the *myl7:DsRed2-NLS* transgene [58]. In agreement with the assessment of Nr2f1a[+] cardiomyocytes, BIO and XAV treatment at the 20s stage respectively produced a surplus and decrease in the number of ACs (*myl7*:DsRed2-NLS[+]/Amhc[+] cardiomyocytes) by 48 hpf (**Fig 4I–4L**). However, we did not detect changes in the number of VCs (**S8A Fig**), similar to what we have reported previously [51,55]. To determine if the surplus ACs from BIO treatment were due to increased proliferation, we labeled embryos with a pulse of EdU at 24 hpf, ~5hrs after BIO treatment began at the 20s stage (**S9A**, **S9B**, **and S9D Fig**). We did not find a difference in proliferation rate of the ACs at this stage, suggesting that increased Wnt signaling beginning at the 20s stage was not increasing cardiomyocyte number via promoting proliferation.

While our analysis of the effects of Tcf7l1a and Wnt signaling on the *3'reg1* reporter, Nr2f1a expression, and AC differentiation are consistent with previous work indicating that Wnt signaling can relieve transcriptional repression from Tcf7l1a [51], we sought to further test this functional relationship of Tcf7l1a to Wnt signaling in this context. To determine of Tcf7l1a functions downstream of canonical Wnt signaling in the repression of the *3'reg1* reporter and differentiation of ACs, we overexpressed Tcf7l1a, by injecting *tcf7l1a* mRNA, and depleted Tcf7l1a while concurrently activating or inhibiting Wnt signaling with BIO and XAV. Consistent with Tcf7l1a functioning as a repressor downstream of Wnt signaling, we found that increasing Tcf7l1a expression was able to suppress the activation of the reporter and increase in the number of ACs by BIO (**S10A**, **S10B**, **S10C**, **S10D**, **S10G**, **S10H**, **S10J**, **S11A**, **S11B**, **S11C**, **S11D**, **S11G**, **S11H**, **and S11J** **Figs**), while concurrently increasing Tcf7l1a and inhibiting Wnt signaling with XAV did not further exacerbate the decrease in reporter expression or loss of ACs (**S10A**, **S10G**, **S10H**, **S10J**, **S11A**, **S11G**, **S11H**, **and S11J**). Concurrent depletion of Tcf7la1 and inhibition of Wnt signaling was able to restore *3'reg1* reporter expression and the decrease the number of ACs in the hearts (**S10A**, **S10I**, **S10J**, **S11A**, **S11I**, **and S11J**), while concurrent depletion of Tcf7l1a and BIO treatment did not further enhance reporter activation or the number of differentiated ACs. Cumulatively, these data suggest that Wnt signaling is necessary and sufficient by the 20s stage to activate the transgenic *3'reg1:GFP* reporter and promote a surplus of differentiated ACs, likely through derepression of Tcf7l1a.

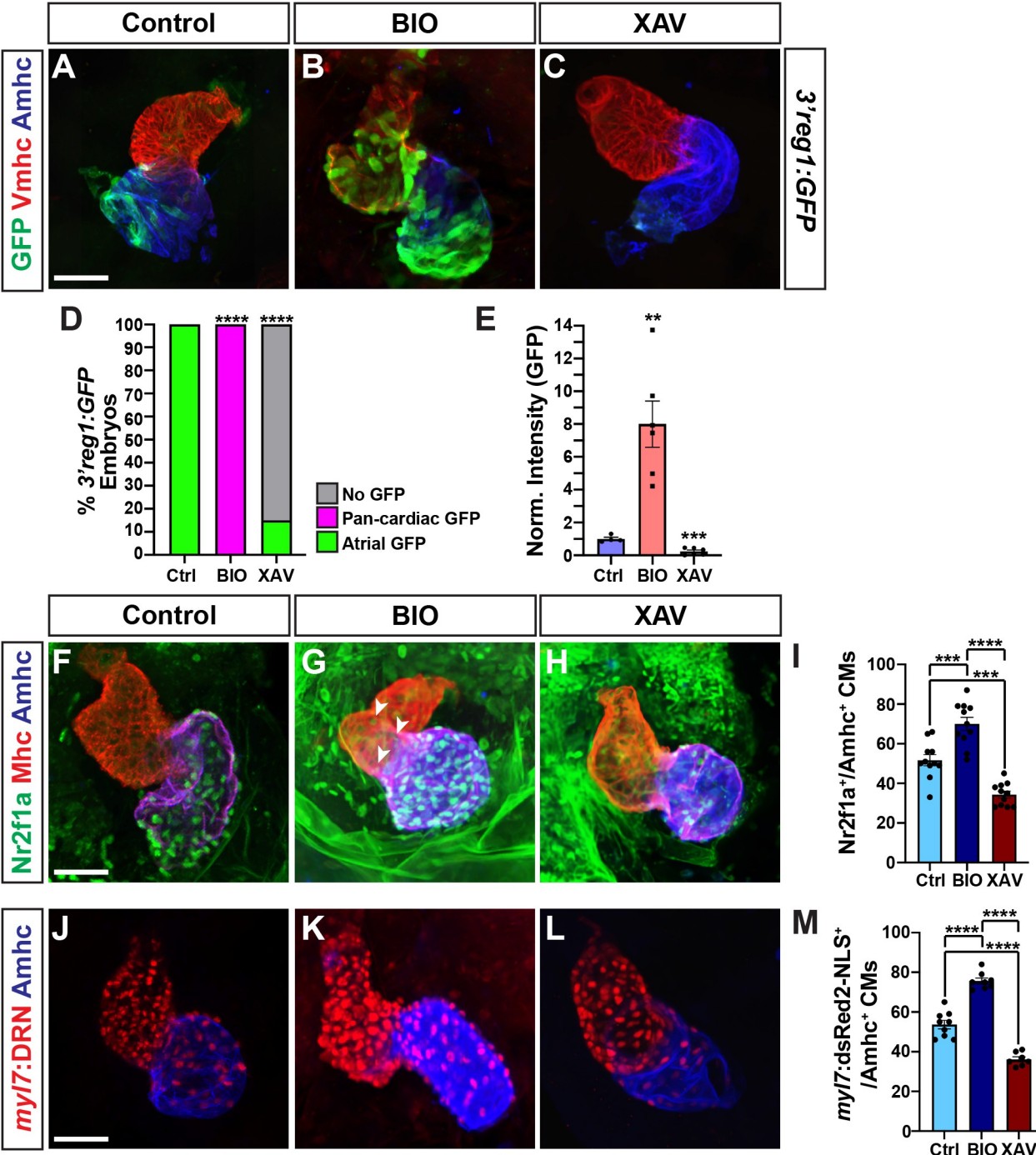

**Fig 4. Wnt signaling promotes *3'reg1* reporter expression and AC differentiation. A-C)** Confocal images of hearts from control, BIO-, and XAV-treated *3'reg1:GFP* embryos stained for *3'reg1*:GFP (green), Vmhc (red), Amhc (blue). **D)** The percentage of control and treated *3'reg1*:GFP embryos with atrial, pan-cardiac, and lacking expression in their hearts. Control (n = 35); BIO (n = 32); XAV (n = 49). **E)** Normalized intensity of GFP expression in hearts of control and treated *3'reg1*:GFP embryos. Control (n = 4); BIO (n = 6); XAV (n = 6). **F-H)** Confocal images of hearts from control, BIO-, and XAV-treated embryos stained for Nr2f1a (green), Mhc (pan-cardiac, red), Amhc (blue). **I)** The number of Nr2f1a$^+$/Amhc$^+$ cardiomyocytes with the hearts of control, BIO-, and XAV-treated embryos. Control (n = 12); BIO (n = 11); XAV (n = 11). **J-L)** Confocal images of hearts from control, BIO-, and XAV-treated *myl7:DsRed2-NLS* (*myl7:DRN*) embryos stained for DsRed-NLS (pan-cardiac nuclei—red) and Amhc (blue). **M)** The number of *myl7*:DsRed2-NLS$^+$/Amhc$^+$ cardiomyocytes (ACs) within the hearts of control, BIO-, and XAV-treated *myl7:DsRed2-NLS* embryos. Control (n = 9); BIO (n = 10); XAV (n = 8). Scale bars: 50 μm. Error bars in graphs indicate s.e.m. ** indicate P < 0.005, *** indicate P < 0.001. **** indicate P < 0.0001.

## Foxf1 promotes Nr2f1a expression in ACs

In mice, Foxf1 and Foxf2 TFs control the proper differentiation of posterior SHF progenitors downstream of Hh signaling [30,59]. Thus, we were intrigued by the conserved Foxf sites within the *3'reg1* enhancer (**Fig 1C**). We found that in transient transgenic embryos deletion or mutation of each of the 3 Foxf binding sites diminished the percentage of embryos with GFP in their atria relative to injection of the wild-type (WT) *3'reg1* reporter (**Fig 5A–5C**), suggesting the putative Foxf sites may be required to promote *3'reg1:GFP* expression. To further test this hypothesis, murine *Foxf1* mRNA was injected into *3'reg1:GFP* transgenic embryos. We found Foxf1 was sufficient to expand *3'reg1:GFP* expression into VCs and promote an

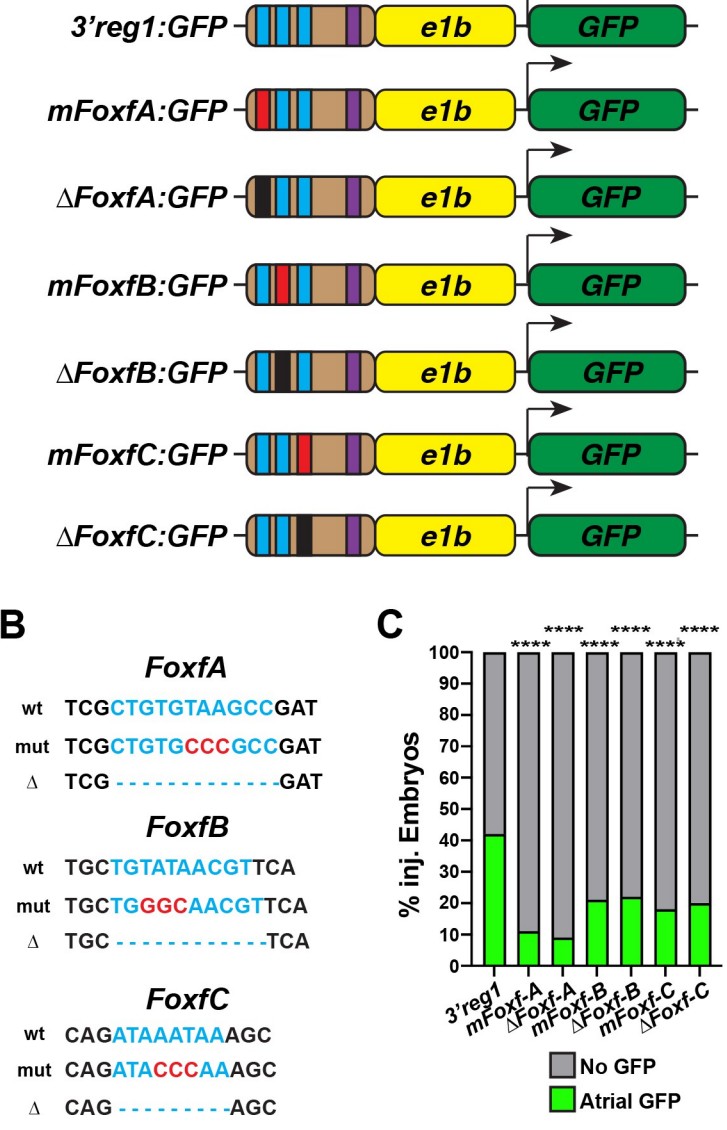

**Fig 5. Foxf1 sites are required for *3'reg1* reporter expression. A)** Schematics of *3'reg1:GFP* reporter constructs with WT, deleted, and mutated Foxf sites. Foxf sites (blue), mutated Foxf sites (red), deleted Foxf sites (black), and Lef/Tcf site (purple). **B)** Sequences of the WT (blue), mutated (red), and deleted (dashes) Foxf sites within the *3'reg1* enhancer. **C)** The percentage of transient transgenic embryos with reporter expression in the atria. Control (*3'reg1*) (n = 150); *mFoxf-A* (n = 60); *ΔFoxf-A* (n = 101); *mFoxf-B* (n = 49); *ΔFoxf-B* (n = 200); *mFoxf-C* (n = 98); *ΔFoxf-C* (n = 195).

increase in intensity of GFP-expressing cardiomyocytes by 48 hpf (**Fig 6A–6E**). Conversely, injecting a dominant-negative murine *Foxf1* (*dnFoxf1*) mRNA into the *3'reg1:GFP* transgenic embryos was able to inhibit reporter expression within the heart (**Fig 6A, 6C, 6D and 6E**). Although zebrafish mutants for *foxf1* have been described [60], if they have cardiac defects was not reported. Thus, we sought to determine if *foxf1* is required for *3'reg1:GFP* reporter expression and we targeted zebrafish *foxf1* using the CRISPR-Cas12a system [61–63] with guides we found were highly efficient in creating deletions (**S12A and S12B Fig**). Although we did not observe overt cardiac defects, injection of the *foxf1* CRISPRs was sufficient to reduce expression of the *3'reg1:GFP* reporter in transgenic embryos (**S12C–S12E Fig**). Given the effects on the *3'reg1:GFP* reporter, we asked if Foxf activity was necessary and sufficient to promote Nr2f1a$^+$ ACs and increase the number of differentiated ACs. We found that embryos injected with *Foxf1* and *dnFoxf1* mRNA respectively showed increases and decreases in *nr2f1a* expression in hearts at 24 hpf, and in the number of Nr2f1a$^+$/Amhc$^+$ ACs, the intensity of Nr2f1a$^+$ cardiomyocytes, and *myl7:DsRed2-NLS$^+$*/Amhc$^+$ cardiomyocytes within their hearts at 48 hpf (**Figs 6F–6M, S7A, S7B and S13A–S13D**), without affecting the number of VCs (**S8B Fig.**). However, unlike with Tcf7l1a depletion and activation of Wnt signaling, we did not find embryos with sporadic VCs expressing Nr2f1a at low levels (**Figs 6F, 6G, S13A, S13B, and S13E**). EdU pulsing at 24 hpf following *Foxf1* mRNA injection also did not show a difference in proliferation rate compared to controls (**S9A, S9C, and S9D Fig**). Thus, our data suggest that Foxf1 can activate the transgenic *3'reg1:GFP* reporter and promote an increase in differentiated Nr2f1a-expressing ACs.

## Hh signaling represses *3'reg1* reporter expression

In mice, Hh signaling has been established as a key regulator of posterior SHF differentiation upstream of factors, including Foxf1 and Foxf2, and Wnt signaling [29–31], and has previously been implicated in regulating Nr2f2 expression in P19 cells [64]. Thus, we asked if Hh signaling could affect the *3'reg1:GFP* reporter within the atria. Transgenic *3'reg1:GFP* embryos were treated with the Hh signaling inhibitor Cyclopamine (CYA) [65] and the Hh signaling activator SAG [66] beginning at the 20s stage, which is after previous reports demonstrated that Hh signaling affects the production of cardiac progenitors [56]. However, in contrast to the effects of Wnt and Foxf1 manipulation, embryos treated with CYA at the 20s stage unexpectedly showed a modest expansion of the *3'reg1:GFP* reporter within the atrium, while SAG-treated *3'reg1:GFP* embryos showed a reduction of *3'reg1*:GFP-expression (**S14A–S14C and S14H Fig**). To confirm that Hh signaling represses *3'reg1:GFP* reporter expression, we examined the *3'reg1:GFP* reporter in *smoothened* (*smo*) mutants. Consistent with CYA treatments, we found a modest expansion of the *3'reg1:GFP* reporter within the atria of *smo* mutants (**S14I–S14K Fig**). However, despite the effects on the *3'reg1:GFP* reporter within the atrium, modulation of Hh signaling did not affect the number of Nr2f1a$^+$ ACs (**S15A–S15C and S15H Fig**). As Hh signaling may be upstream of Wnt signaling in posterior SHF differentiation in mammals [31,67], we also concurrently manipulated Wnt and Hh signaling at the 20s stage with the aforementioned drugs. We found that simultaneously inhibiting Wnt and Hh signaling and activating Wnt and Hh signaling, respectively, decreased and increased expression of the *3'reg1:GFP* reporter and the number of Nr2f1$^+$ ACs within the hearts, similar to inhibition or activation of Wnt signaling alone (**S14A–S14H and S15A–S15H Figs**). Therefore, our data suggest that in contrast to mammalian venous pole differentiation Hh signaling is likely not affecting AC differentiation at these stages of cardiogenesis nor is it promoting Nr2f expression in ACs upstream of Wnt or Foxf1 in zebrafish embryos.

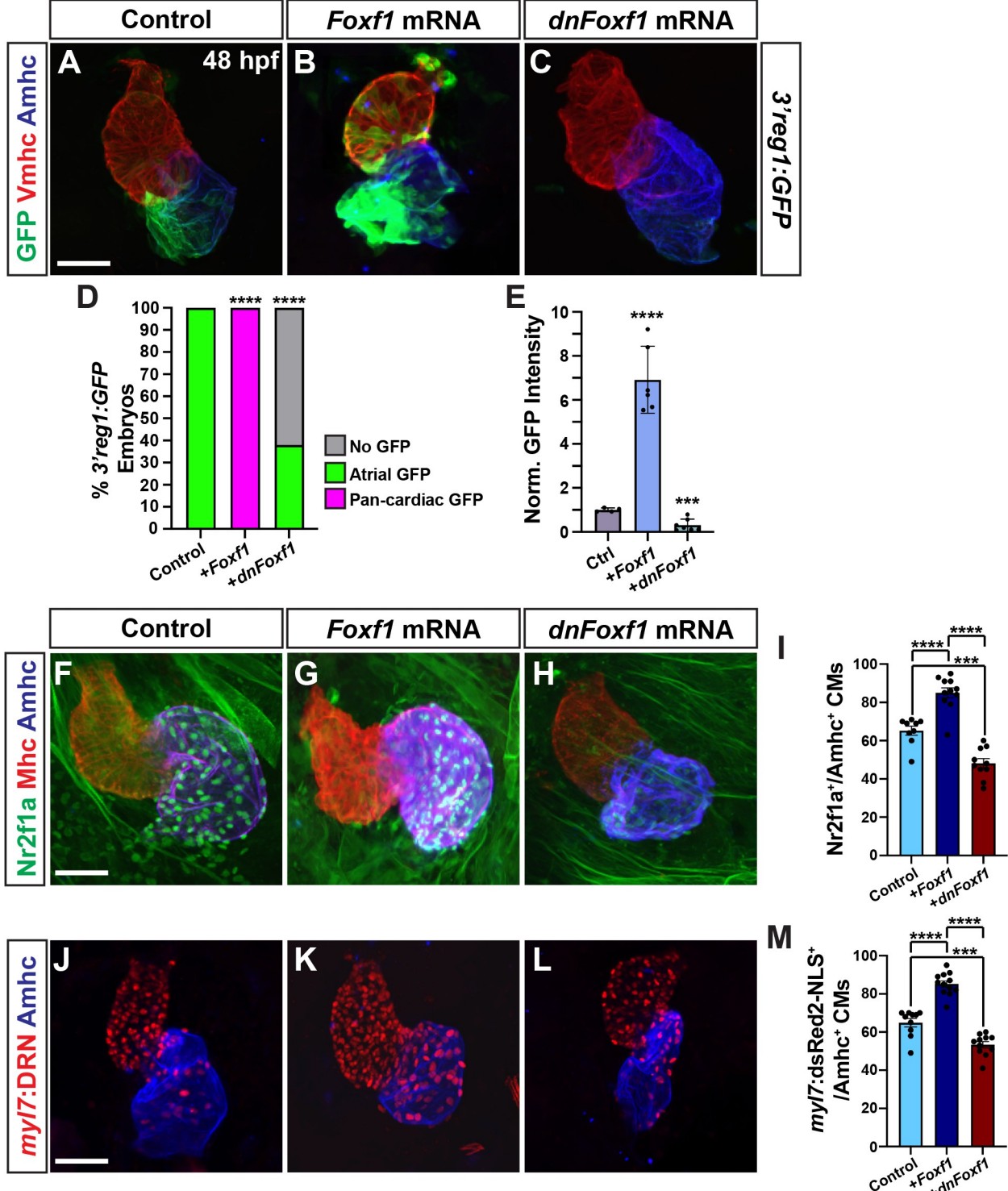

**Fig 6. Foxf1 promotes *3'reg1* reporter expression within the heart. A-C)** Confocal images of hearts from control uninjected, *Foxf1* mRNA, and *dnFoxf1* mRNA injected *3'reg1:GFP* embryos stained for *3'reg1*:GFP (green), Vmhc (red), Amhc (blue). **D)** The percentage of injected *3'reg1:GFP* embryos with atrial expression, pan-cardiac expression, and lacking expression. Control (n = 35); *Foxf1* mRNA (n = 29); *dnFoxf1* mRNA (n = 59). **E)** Normalized intensity of GFP expression in hearts of injected *3'reg1:GFP* embryos. Control (n = 4); *Foxf1* mRNA (n = 6); *dnFoxf1* mRNA (n = 7). **F-H)** Confocal images of hearts from control, *Foxf1* mRNA, and *dnFoxf1* mRNA injected embryos stained for Nr2f1a (green), Mhc (red), Amhc (blue). **I)** The number of Nr2f1a⁺/Amhc⁺ cardiomyocytes (ACs) within the hearts of control, *Foxf1* mRNA, and *dnFoxf1* injected embryos. Control (n = 8); *Foxf1* (n = 11); *dnFoxf1* (n = 8). **J-L)** Confocal images of hearts from control, *Foxf1* mRNA, and *dnFoxf1* injected *myl7:DsRed2-NLS* (*myl7:*

*DRN*) embryos stained for DsRed-NLS (red) and Amhc (blue). **M)** Quantification of the number of *myl7*:DsRed2-NLS⁺/Amhc⁺ cardiomyocytes (ACs) within the hearts of control, *Foxf1* mRNA, and *dnFoxf1* injected *myl7:DsRed2-NLS* embryos. Control (n = 10); *Foxf1* (n = 13); *dnFoxf1* (n = 11). Scale bars: 50 μm. Error bars in graphs indicate s.e.m. *** indicate P < 0.001. **** indicate P < 0.0001.

## Wnt signaling functions downstream of Foxf1

To decipher the functional relationship between Wnt signaling/Tcf7l1a and Foxf1 on activation of the *3'reg1*:*GFP* reporter within the atrium, we mutated each of the Foxf1 sites and all 3 of the Foxf1 sites in combination with mutating the Tcf site in the *3'reg1*:*GFP* reporter construct (**S16A Fig**). We found that mutations of any of the Foxf1 sites along with the Tcf site was sufficient to reduce the percentage of injected transient transgenic embryos with atrial expression and prevent the expansion of expression into the VCs found with mutation of the Tcf site alone (**S16B Fig**). Mutation of all three of the sites along with the Tcf site did not produce any expression in injected embryos (**S16B Fig**). Thus, in the transient transgenic embryos, the Foxf1 sites are needed within the *3'reg1* reporter for the ectopic expression induced by mutation of the Tcf site.

We next simultaneously manipulated these signals within the transgenic *3'reg1*:*GFP* embryos. While *3'reg1*:*GFP* embryos injected with *Foxf1* mRNA had pan-cardiac expression, the majority of *Foxf1* mRNA-injected *3'reg1*:*GFP* embryos treated with XAV at the 20s stage lose *3'reg1*:*GFP* reporter expression within the atrium by 48 hpf, similar to XAV treatments alone (**Fig 7A–7E**). We next simultaneously injected *Foxf1* and zebrafish *tcf7l1a* mRNAs into the *3'reg1*:*GFP* embryos. Consistent with the Tcf7l1a functioning as a repressor, *tcf7l1a* overexpression abrogated *3'reg1*:*GFP* reporter expression within the atrium (**Fig 7F, 7H, and 7J**). When both TFs were co-overexpressed in *3'reg1*:*GFP* embryos, we did not observe pan-cardiac expression of the *3'reg1*:*GFP* reporter within the atrium at 48 hpf (**Fig 7F–7J**). Next, we co-injected the *tcf7l1a* MO and *dnFoxf1* mRNA into the *3'reg1*:*GFP* embryos. In contrast with the simultaneous mutation of Foxf1 and Tcf sites in transient transgenic embryos, we found that the *dnFoxf1* was not sufficient to block the pan-cardiac expression induced by Tcf7l1a depletion in the stable *3'reg1*:*GFP* embryos (**Fig 7K–7O**). Since these data suggested that Wnt signaling may function downstream of Foxf1, we asked if Foxf1 was sufficient to promote an increase in Wnt signaling within embryos. RT-qPCR on cDNA from isolated hearts of 24 hpf embryos injected with *Foxf1* mRNA promoted increases in the expression of *axin1* and *axin2*, Wnt signaling-responsive genes that negatively regulates canonical Wnt signaling [68–70], and a decrease in the expression of the Wnt signaling inhibitor *dkk1a* [68–70] (**S17A–S17C Fig**). Conversely, injection of *dnFoxf1* mRNA inhibited the expression of *axin1* and *axin2*, and promoted *dkk1a* expression in the hearts of 24 hpf embryos (**S17A–S17C Fig**). Additionally, we injected *Foxf1* and *dnFoxf1* mRNAs into embryos of the *Tg(7xTcf-Xla.Sia:GFP)* canonical Wnt signaling reporter line [71]. Consistent with Foxf1 promoting an increase in Wnt signaling, *Foxf1* mRNA injection was able to induce expression of the *7xTcf-Xla.Sia*:GFP in and adjacent to the heart at 34 hpf, which is prior to its expression in the heart [71], while dnFoxf1 mRNA reduced expression of the reporter (**S18A–S18D Fig**). Altogether, these data suggest that Wnt signaling functions downstream, or parallel, to Foxf1 in the activation of the *3'reg1*: *GFP* reporter.

## Wnt signaling and Foxf1 require Nr2f1a to promote surplus ACs

Nr2f1a is required for AC differentiation within the early zebrafish heart [12]. As our results support that enhanced Wnt signaling and Foxf1 are sufficient to promote more differentiated ACs, we asked if this ability was dependent on Nr2f1a. Embryos resulting from crosses of

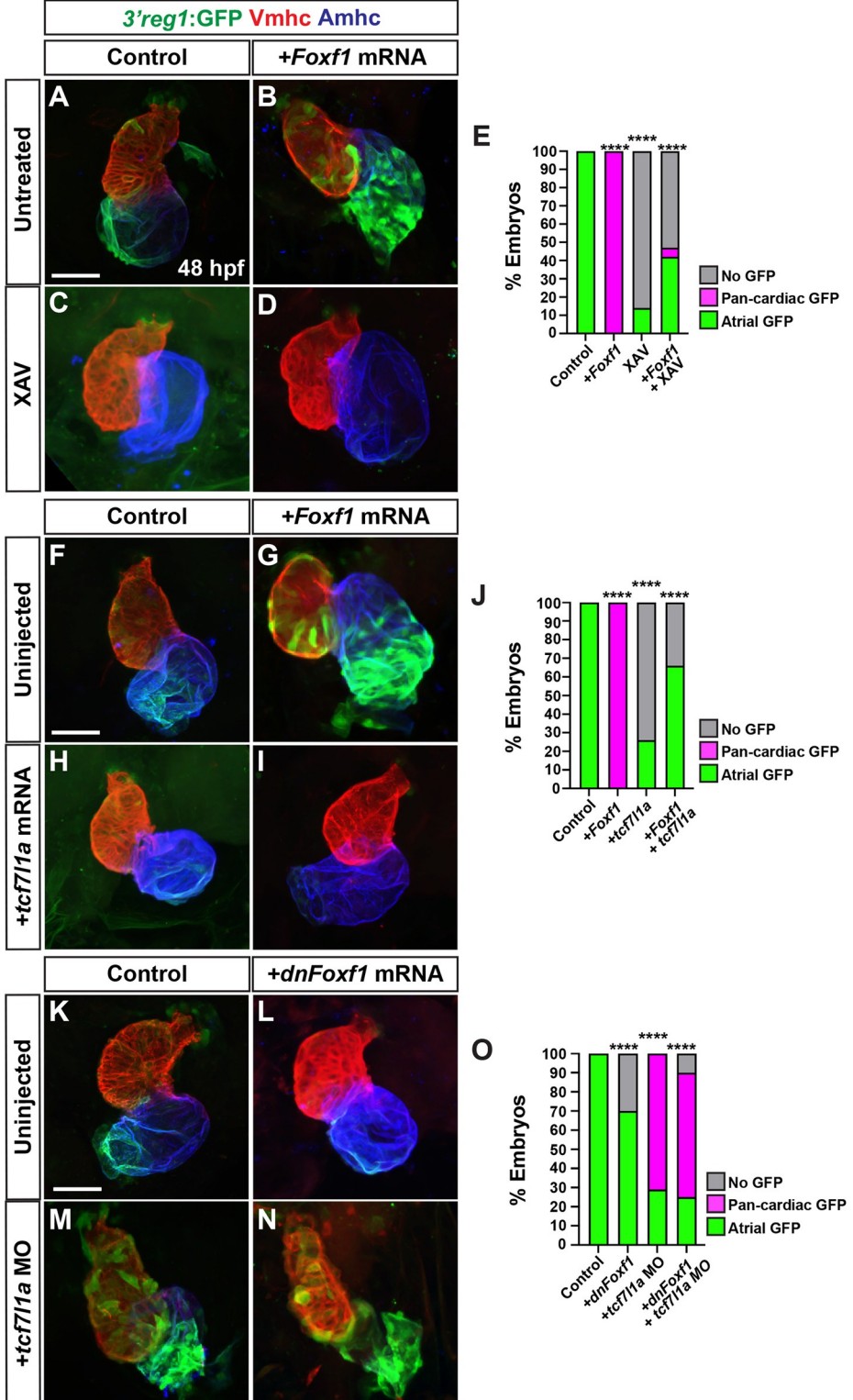

**Fig 7. Tcf7l1a limits the ability of Foxf1 to promote *3'reg1* reporter expression. A-D)** Confocal images of hearts from control, *Foxf1* mRNA injected, XAV-treated, and *Foxf1* mRNA injected plus XAV-treated *3'reg1:GFP* embryos. **E)** The percentage of control, *Foxf1* mRNA injected, XAV-treated, and *Foxf1* mRNA injected plus XAV-treated *3'reg1: GFP* embryos with atrial expression, pan-cardiac expression, and lacking expression. Control (n = 26); *Foxf1* (n = 24); XAV n = 98; *Foxf1*-XAV (n = 103). **F-I)** Confocal images of hearts from control, *Foxf1* mRNA injected, *tcf7l1a* mRNA

injected, and *Foxf1* mRNA and t*cf7l1a* mRNA co-injected *3'reg1:GFP* embryos. **J)** The percentage of control, *Foxf1* mRNA injected, *tcf7l1a* mRNA injected, and *Foxf1* mRNA and t*cf7l1a* mRNA co-injected *3'reg1:GFP* embryos with atrial expression, pan-cardiac expression, and lacking expression. Control (n = 39), *Foxf1* (n = 34); *tcf7l1a* (n = 95); *Foxf1-tcf7l1a* (n = 51). **K-N)** Confocal images of hearts from control, *dnFoxf1* mRNA injected, *tcf7l1a* MO injected, and *Foxf1* mRNA and t*cf7l1a* MO co-injected *3'reg1:GFP* embryos. **O)** The percentage of control, *dnFoxf1* mRNA injected, *tcf7l1a* MO injected, and *Foxf1* mRNA and t*cf7l1a* MO *3'reg1:GFP* embryos with atrial expression, pan-cardiac expression, and lacking expression. Control (n = 27), *dnFoxf1* (n = 102); *Tcf7l1a-MO* (n = 96); *dnFoxf1-tcf7l1aMO* (n = 100). Hearts are stained for *3'reg1*:GFP (green), Vmhc (red), Amhc (blue)Scale bars: 50 μm. **** indicate P < 0.0001.

*nr2f1a*$^{+/-}$ fish that carry the *myl7:DsRed2-NLS* transgene were treated with BIO at the 20s stage. We found that BIO treatment was not sufficient to produce an increase of ACs in 48 hpf *nr2f1a* mutants, while it was sufficient to produce an increase in the number of ACs in their WT siblings (**Fig 8A–8E**). Similarly, injection of *Foxf1* mRNA into embryos resulting from crosses of *nr2f1a*$^{+/-}$ fish that carry the *myl7:DsRed2-NLS* transgene showed that *Foxf1* mRNA was no longer sufficient to produce a surplus of ACs in *nr2f1a* mutants, unlike their control sibling embryos (**Fig 8F-8J**). Thus, our data support that Nr2f1a is required for Wnt signaling and Foxf1 to be sufficient to promote a surplus of ACs.

## Wnt signaling and Foxf1 require *3'reg1* to promote ACs

Since the *3'reg1:GFP* reporter is expressed in ACs, we wanted to determine if the endogenous *3'reg1* enhancer is required *in vivo* to promote or maintain Nr2f1a expression in ACs. Therefore, we deleted the endogenous *3'reg1* enhancer in zebrafish using a CRISPR-Cas12a system [72] (**Figs 9, S19A, and S19B**). We found that embryos with homozygous deletion of the endogenous *3'reg1* enhancer overtly did not show defects (**S9C and S9D Fig**). Although we did not find a difference in the number of ACs (Nr2f1a$^+$/Amhc$^+$ cardiomyocytes) in the hearts of *3'reg1* deletion mutants compared to their WT sibling embryos at 48 hpf (**Fig 9B, 9C, 9F, 9G, 9H, and 9K**), quantification of Nr2f1a staining intensity in the nuclei appeared to be reduced relative to WT sibling embryos (**S20A–S20C Fig**). Thus, our data support that *in vivo* deletion of the *3'reg1* enhancer may lead to a reduction in the levels of Nr2f1a. However, this reduction is not sufficient enough to recapitulate atrial defects observed in *nr2f1a* mutants. Consequently, we then asked if Wnt signaling and Foxf1 require the endogenous *3'reg1* enhancer to be able to produce a surplus of Nr2f1a$^+$ ACs. We found that both BIO treatment and *Foxf1* mRNA injection were unable to produce a surplus of Nr2f1a$^+$/Amhc$^+$ cardiomyocytes within the hearts of homozygous *3'reg1* deletion mutants (**Fig 9B–9K**). Overall, these data suggest that the endogenous *3'reg1* is primarily required to mediate the sensitivity of Nr2f1a and AC differentiation to Wnt signaling and Foxf1 in the zebrafish atrium, although it may modestly augment the expression of Nr2f1a.

## Discussion

In the present study, our data show that in zebrafish embryos Wnt signaling and Foxf1 function upstream of Nr2f1a in promoting AC differentiation via regulating a conserved *Nr2f1* enhancer, which is located ~2.5kb 3' to zebrafish *nr2f1a* (**Fig 10**). Previous work has placed both these signals downstream of Hh signaling in a regulatory network that controls the differentiation of ACs from the posterior SHF in mice [31,32,67]. Thus, our data support that there are conserved roles for these downstream factors in promoting the proper number of differentiated ACs in the hearts of vertebrates. Numerous studies have investigated the different roles of canonical Wnt signaling in cardiac development in mice, zebrafish, and stem cells [32,33,47,55,73]. Furthermore, our previous study examining the temporal requirements of

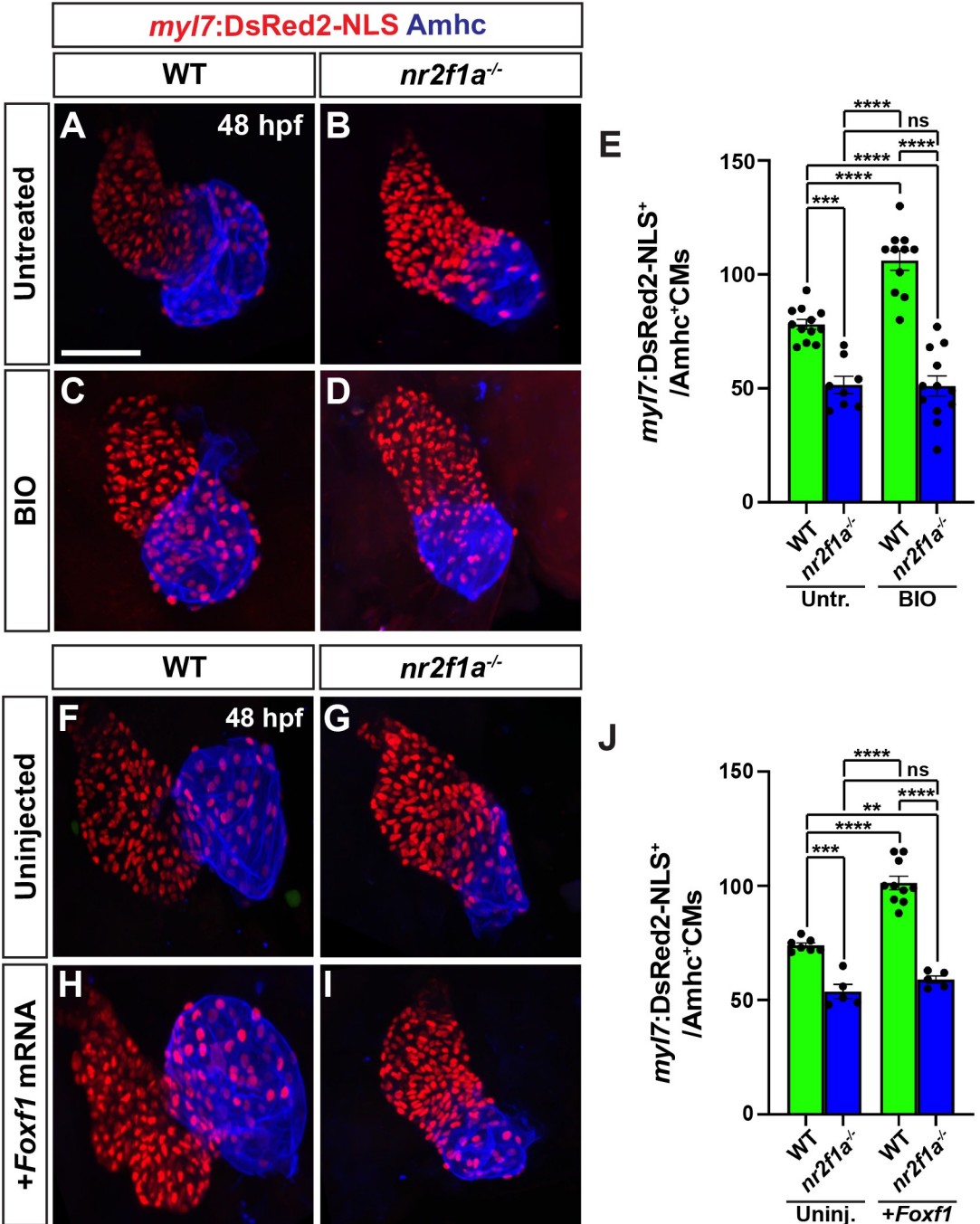

**Fig 8. Wnt signaling and Foxf1 require Nr2f1a to promote a surplus of cardiomyocytes. A-D)** Confocal images of hearts from untreated and BIO-treated WT and *nr2f1a*[-/-] *myl7:DsRed2-NLS* embryos. **E)** The number of *myl7*:DsRed2-NLS[+]/Amhc[+] cardiomyocytes in hearts of untreated and BIO-treated WT and *nr2f1a*[-/-] *myl7:DsRed2-NLS* embryos. Untreated WT (n = 12); untreated *nr2f1a*[-/-] (n = 7); treated WT (n = 11); treated *nr2f1a*[-/-] (n = 10). **F-I)** Confocal images of hearts from uninjected and *Foxf1* mRNA-injected WT and *nr2f1a*[-/-] *myl7:DsRed2-NLS* embryos. **J)** The number of *myl7*:DsRed2-NLS[+]/Amhc[+] cardiomyocytes in uninjected and *Foxf1* mRNA-injected WT and *nr2f1a*[-/-] *myl7:DsRed2-NLS* embryos. Uninjected WT (n = 7); Uninjected *nr2f1a*[-/-] (n = 5); Inj. WT (n = 10); Injected *nr2f1a*[-/-] (n = 5). Hearts are stained for DsRed-NLS (red) and Amhc (blue). Scale bars: 50 μm. Error bars in graphs indicate s.e.m. ** indicate P < 0.01. **** indicate P < 0.001. **** indicate P < 0.0001. ns indicates not a statistically significant difference.

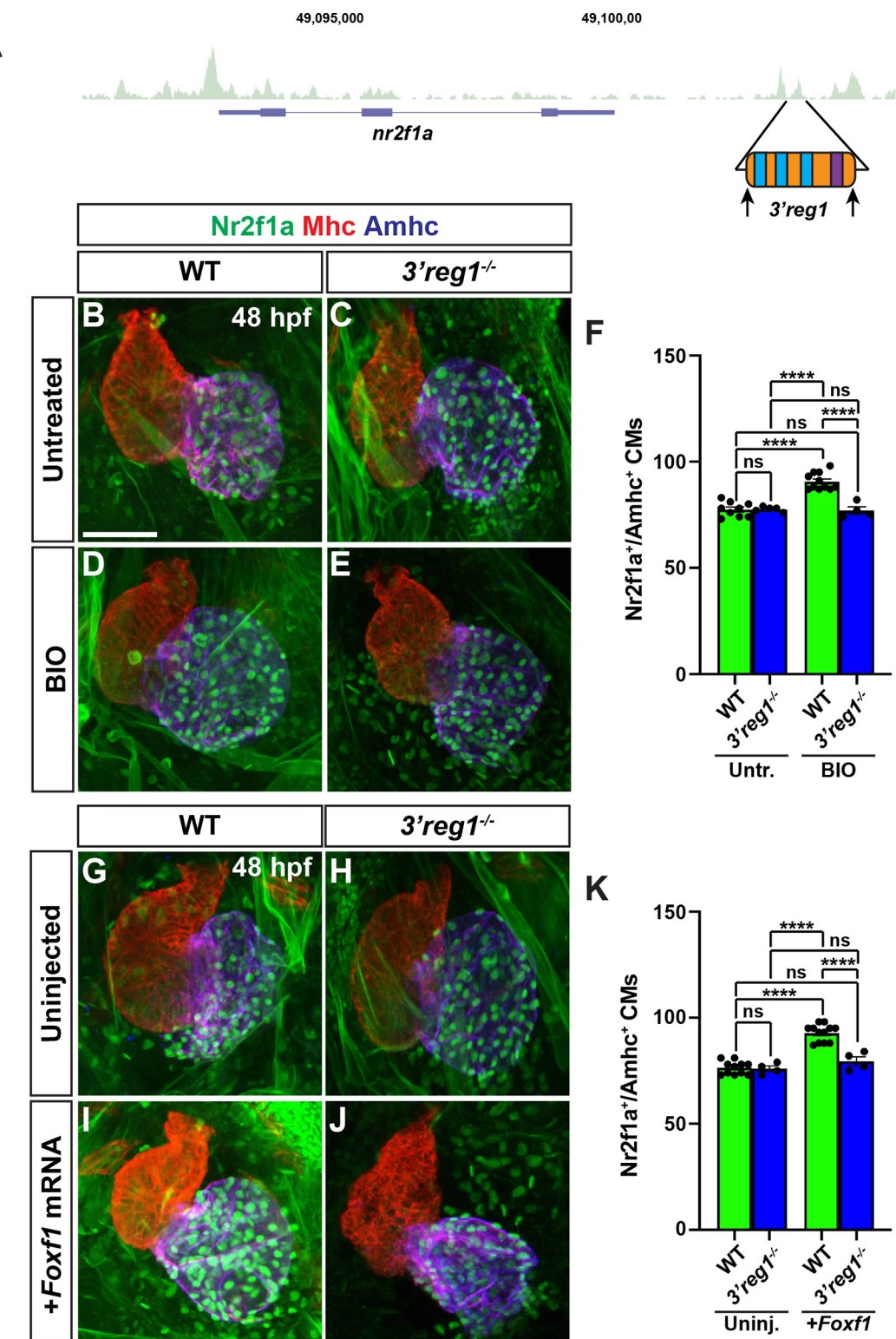

**Fig 9. Wnt signaling and Foxf1 require *3'reg1* to promote a surplus ACs. A)** Schematic of *3'reg1* enhancer deletion generated using CRISPR/Cas12a. Arrows indicate location of the guides. **B-E)** Confocal images of hearts from untreated and BIO-treated WT and *3'reg1^Δ/Δ^* embryos stained for Nr2f1a (green), Mhc (red), and Amhc (blue). BIO was unable to promote an increase in ACs in *3'reg1^-/-^* embryos. **F)** The number of Nr2f1a⁺/Amhc⁺ cardiomyocytes in untreated and BIO-treated WT and *3'reg1^-/-^* embryos. Untreated WT (n = 9); untreated *3'reg1^-/-^* (n = 5); treated WT (n = 11); treated *3'reg1^-/-^* (n = 4). **G-J)**

Confocal images of hearts from uninjected and *Foxf1* mRNA-injected WT and *3'reg1*[-/-] embryos stained for Nr2f1a (green), Mhc (red), and Amhc (blue). **K)** The number of Nr2f1a[+]/Amhc[+] cardiomyocytes in uninjected and *Foxf1* mRNA-injected WT and Nr2f1a[+] *3'reg1*[-/-] embryos. Uninjected WT (n = 11); Uninjected *3'reg1*[-/-] (n = 4); Injected WT (n = 12); Injected *3'reg1*[-/-] (n = 4). Scale bar: 50 μm. Error bars in graphs indicate s.e.m. **** indicate P < 0.0001. ns indicates not a statistically significant difference.

Wnt signaling in zebrafish showed that it is necessary and sufficient to specifically promote surplus ACs during later somitogenesis immediately prior to the formation of the heart tube [51,55], a time point when posterior SHF progenitors are adding to the zebrafish heart [74]. Additional work in zebrafish has shown that Wnt signaling at the lateral borders of the heart field and at the venous pole is required for the differentiation of pacemaker cardiomyocytes [53]. Our present study corroborates the role for Wnt signaling in promoting the proper number of ACs as the heart tube is forming during late somitogenesis, which we propose is through affecting the differentiation of ACs from the zebrafish posterior SHF. Furthermore, our work supports that the ability of Wnt signaling to promote ACs necessitates Nr2f1a, whose expression is regulated via the derepression of Tcf7l1a.

Despite the established roles of Foxf1/2 TFs in regulating differentiation of the posterior SHF in mice [59,75,76], if Foxf TFs play similar roles in zebrafish heart development has not been investigated. Previous work examining Foxf factors in zebrafish demonstrated redundant requirements for Foxf1/2 TFs in craniofacial development [60,77]. Similar to the roles of Foxf1/2 in mice [30,59,75], our data suggest that the Foxf TFs are necessary and sufficient to promote the formation of ACs in zebrafish embryos. Although our data did not yet specifically address the temporal or spatial requirements of Foxf factors, the necessity of Nr2f1a in promoting the proper number of differentiated ACs would also support a model that affects the differentiation of zebrafish posterior SHF progenitors. Thus, as with Wnt signaling, our data support a conserved requirement for Foxf TFs upstream of Nr2f1a in controlling the differentiation of ACs at the venous pole in vertebrates. However, although we found that depletion of *foxf1* affected the expression of the *3'reg1:GFP* reporter, zebrafish *foxf1* crispants did not overtly have cardiac defects. Additionally, both Wnt signaling and Foxf1 could induce ectopic *3'reg1:GFP* reporter in VCs. However, Tcf7l1a depletion and Wnt signaling were able to induce

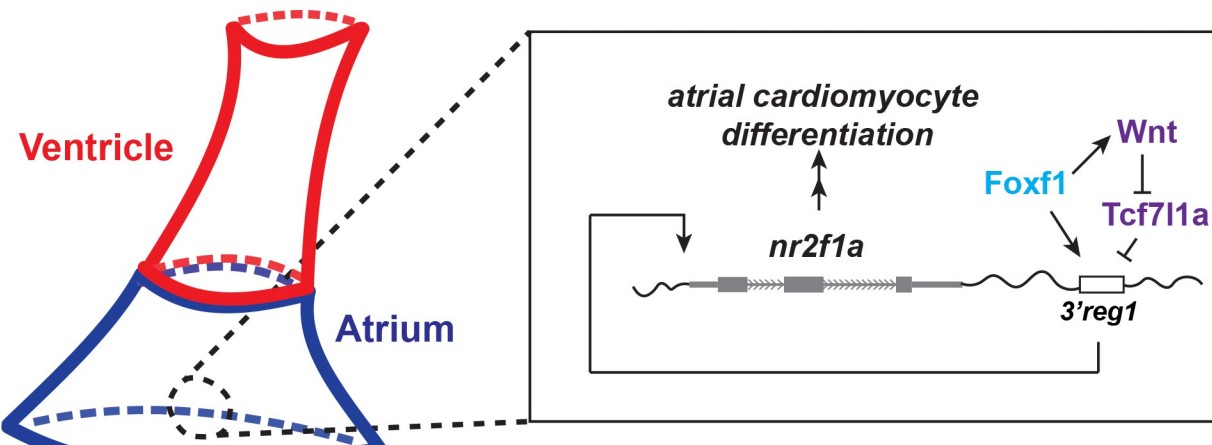

**Fig 10. A Wnt-Foxf1-Nr2f1a cascade directs atrial cardiomyocyte differentiation.** Model depicting the Foxf1 and Wnt signaling-dependent regulatory network that controls Nr2f1a expression and the role of the *3'reg1* enhancer in the atrial cardiomyocyte differentiation during early heart tube development.

a few ectopic VCs with low levels of Nr2f1a expression, which was not observed with Foxf1. Hence, there are differences between the sensitivity of the *3'reg1* reporter and induction of endogenous Nr2f1a to manipulation of Wnt signaling and Foxf factors. A limitation of this type of analysis is the lack of input from additional CREs that may influence endogenous gene expression. We postulate that these differences are due to a lack of input from additional CREs and that multiple *foxf* factors in zebrafish likely function redundantly to promote AC differentiation of zebrafish, as with Foxf1/2 in mice [59,75], which will be investigated more in the future.

Integrating how Wnt signaling and Foxf TFs function upstream of Nr2f1a, our analysis supports that Wnt signaling, via derepression of Tcf7l1a, functions downstream, or at the level, of Foxf1. This functional relationship is reminiscent of other developmental contexts, such as osteogenesis [78,79], where Foxf factors have been found to regulate the canonical Wnt signaling pathway [80]. Despite their functional relationship in promoting the differentiation of zebrafish ACs, we have not determined the specific mechanism by which Foxf factors regulate Wnt signaling in this context. Given the adjacent Foxf sites and Lef/Tcf site within the *3'reg1* CRE, one might predict that Foxf factors, which can act as pioneer factors [81,82], facilitate the derepression of Tcf7l1a via making the CRE accessible to Wnt signaling. However, our data with the stable transgenic reporter would argue against this model, as we find Tcf7l1a does not appear to require Foxf for the derepression. In contrast to the epistasis in the stable reporter, our analysis of the combinatorial mutations of the Foxf and Tcf sites in transient transgenic embryos suggests that the Foxf sites are required for Tcf-dependent activation of the reporter. Thus, another possibility is that Foxf reinforces the activation of the CRE via simultaneously acting on the *3'reg1* enhancer and independently promoting Wnt signaling via derepression of Tcf7l1a. While Foxf1 can function as a transcriptional activator [83], it is predominantly known as a transcriptional repressor [84,85]. The dependence of the *3'reg1*:*GFP* reporter on the putative Foxf sites would minimally suggest that in this context these sites are being utilized by a transcriptional activator, which correlates with the evidence that Foxf1 is able to promote *3'reg1*:*GFP* reporter expression and atrial differentiation. Nevertheless, we acknowledge that in the absence of evidence for direct binding to the endogenous *3'reg1* enhancer, multiple other scenarios are plausible, including that Foxf1 represses a Wnt inhibitor. However, in mice, in addition to repression of factors that limit differentiation of the posterior SHF, Foxf1/2 are also required for the activation of chamber-specific differentiation programs [30] which could also suggest that it has dual functions as a repressor and activator in this context.

In mammals, Hh signaling sits atop a regulatory network that controls the proper timing of differentiation from posterior SHF progenitors through Tbx5 and Foxf TFs, and Wnt and RA signaling [30,32,73]. Although our data support that downstream factors in this regulatory network, Wnt signaling and Foxf TFs, are conserved in regulating posterior SHF addition in zebrafish, our data and previous studies do not support the conservation of the upstream factors [30,31]. Previous work established that in zebrafish Hh signaling is required during early patterning stages to promote the proper number of differentiated ACs and VCs [56], while a requirement at later stages of somitogenesis and posterior SHF addition equivalent to those examined in this study was not found. Consistent with the previous observations of the temporal requirements of Hh signaling [56,86], we did not find it was required upstream of Wnt signaling or Foxf. Instead, we found that there was a minimal effect of Hh signaling repressing the *3'reg1*:*GFP* reporter, which was the opposite of what would be expected from the mammalian data. Furthermore, we did not find evidence that Hh regulates Nr2f1a or affects the number of differentiated ACs in zebrafish during late somitogenesis and early heart tube formation. Previous work has also established that Tbx5, which functions downstream of Hh and RA signaling in mice in posterior SHF differentiation [30–32], is not required for zebrafish atrial differentiation [87–89]. It is interesting to consider the lack of conservation of these

upstream factors in the regulation of the vertebrate posterior SHF. The posterior SHF-derived cells that contribute to the single atrium and venous pole in zebrafish are significantly less than that found in mice [74,86]. By virtue of its lack of requirement, our work would support the proposed hypothesis whereby heterochrony of a Hh-dependent regulatory network may have contributed to the enlargement of the posterior SHF and the concurrent the evolution of the pulmonary system in air-breathing vertebrates [73,86].

Specifically evaluating the *3'reg1* enhancer, our data support that it is highly conserved among vertebrates, but is absent in agnathans (hagfish and lamprey) and birds. A lack of this CRE in agnathans could suggest that it evolved beginning in cartilaginous fishes. Although not explored here, *Nr2f2* genes, which diverged from *Nr2f1* early in the vertebrate lineage [37], also have a conserved putative enhancer located in a similar region (**S21 Fig**), suggesting these enhancers may have had a similar origin at least in agnathans prior to the divergence of these paralogs. Although the *Nr2f1 3'reg1* is lost in birds, the conservation of the putative *Nr2f2-3'reg1* within birds could reflect the greater reliance of *Nr2f2* within their hearts similar to its role in mammals. The general conservation of the *3'reg1* in vertebrates is interesting in light that it has been suggested that CREs regulating cardiac expression are actually poorly conserved in vertebrates [90]. Additionally, deletion of the *3'reg1* did not cause overt loss of or failure to maintain Nr2f1a and ACs deficits, although the expression was potentially reduced. Thus, other CREs are likely involved in or compensate for the loss of the *3'reg1*, which is not uncommon [91], and may also be a reason its loss could be tolerated in agnathans and birds. We focused on the *3'reg1* because of the functional importance of *nr2f1a* in zebrafish atrial differentiation and that the transgenic *3'reg1*:*GFP* reporter recapitulated aspects of Nr2f1a expression within the atrium. Our data support that Wnt signaling and Foxf factors require the endogenous *3'reg1* enhancer to be able to augment the number of ACs, even if the element itself is not required to promote or maintain normal Nr2f1a expression within the heart. Furthermore, although this was the only conserved *Nr2f1* CRE thus far that we identified promotes AC-specific expression, we did find nearby CREs that promoted expression more broadly in cardiomyocytes. We have not yet examined the coordination of these CREs and if expression driven with the region 3' to *nr2f1a* encompassing all the CREs better reflects endogenous *nr2f1a* within ACs. Additionally, as the transgenic *3'reg1* reporter expression initiates at early heart tube stages, which is later than endogenous *nr2f1a* expression occurs in the anterior lateral plate mesoderm during early somitogenesis, and demonstrates a graded pattern, this would suggest that this enhancer is required to promote or maintain *nr2f1a* expression within more venous ACs at later stages, and that other CREs likely provide input that fully recapitulates its expression in atrial cardiomyocytes.

In summary, our data have provided insights into a Foxf-Wnt-Nr2f regulatory network that determines the number of differentiated ACs within the zebrafish heart. Our study has implications for our understanding of the development and evolution of gene regulatory networks controlling posterior SHF and AC differentiation in vertebrates and given the conserved requirements of Nr2f factors could be used to inform us of mechanisms underlying congenital heart defects that affect the atria in mammals. Future studies will be aimed at deciphering the specific transcriptional mechanisms that control the logic of this regulatory network and expanding our analysis to additional CREs directing *Nr2f1* and *Nr2f2* expression in ACs.

## Materials and methods

### Ethics statement

All zebrafish husbandry and experiments were performed following protocols approved by the Institutional Animal Care and Use Committee (IACUC) of Cincinnati Children's Hospital Medical Center (Protocol IACUC2023-1048).

## Zebrafish husbandry and lines used

Adult zebrafish were raised and maintained under standard laboratory conditions [92]. WT fish used were mixed AB-TL background. Transgenic and mutant lines used were: *Tg (-1.5_nr2f1a:GFP)$^{ci1020}$, Tg(-1.4_nr2f1a:GFP)$^{ci1021}$, Tg(-0.7s_nr2f1a:GFP)$^{ci1022}$, Tg (-0.7m_nr2f1a:GFP)$^{ci1023}$, Tg(-0.7l_nr2f1a:GFP)$^{ci1024}$, Tg(5'reg1-nr2f1a:GFP)$^{ci1025}$, Tg (3'reg1-nr2f1a:GFP)$^{ci1026}$, Tg(3'reg2-nr2f1a:GFP)$^{ci1027}$, Tg(3'reg3-nr2f1a:GFP)$^{ci1028}$, Tg (3'reg4-nr2f1a:GFP)$^{ci1029}$, Tg(−5.1myl7:DsRed2-NLS)$^{f2}$* [58], *Tg(-5.1myl7:EGFP)$^{twu26}$* [93], *smo$^{s294}$* [94], *nr2f1a$^{ci1009}$* [35], and *Tg(7xTcf.Xla.Sia:GFP)$^{ia4}$* [71].

## Analysis of *Nr2f* loci and TFBS analysis

ATAC-seq data used in this study on flow-sorted zebrafish ACs at 48 hpf and alignments shown were reported previously (GEO, Accession # GSE 194054) [35]. Regions of open chromatin 700 kb upstream (a gene desert) and 18 kb downstream of the *nr2f1a* zebrafish locus were initially examined within the UCSC browser (https://genome.ucsc.edu). Open chromatin in these regions of the zebrafish genome adjacent to the *nr2f1a* locus that appeared to have conservation based on alignments with genomes available in the UCSC genome browser were selected for additional analysis of conservation through alignments using mVista (http://genome.lbl.gov/vista/mvista/submit.shtml) and subsequent reciprocal nBLASTs on Ensembl (Ensembl.org). Specific sequences from species corresponding to the conserved regions of open chromatin flanking *Nr2f1* loci in other species were retrieved from Ensembl: Australian ghostshark (*Callorhinchus_milii*-6.1.3), Tropical clawed frog (UCB_*Xtro*_10.0), Green anole (*AnoCar*2.0v2), Chicken (bGalGal1.mat.broiler.GRCg7b), House mouse (GRCm39), and Human (GRCh38.p14). The conserved regions in the different species, corresponding to the open chromatin in the zebrafish ACs, were manually aligned using Clustal Omega [95]. Sequences with a high degree of conservation from the alignment were analyzed for the presence of conserved TFBS using CIS-BP [40,41], TomTom [42,43], and JASPAR databases [44].

## Cloning of plasmids

The *nr2f1a* promoter and enhancer reporters were generated using established Gateway cloning methods [96]. The designated regions of the *nr2f1a* promoter were cloned into *pDONR P4-P1r* to generate 5'-entry clones. Subsequently, each of the *p5E-nr2f1a* promoter clones were combined with *pME-EGFP* middle-entry and *p3E-polyA* 3'-entry clones into the previously reported *pDestTol2p2a-cry:DsRed* destination vector [96] to generate the *nr2f1a* promoter reporter constructs. The putative enhancer regions were cloned into the *pE1b-GFP-Tol2-Gateway* vector (Addgene, plasmid # 37846) [38]. A QuickChange Site-Directed Mutagenesis Kit (Agilent, Cat # 200524) was employed to generate the designated deletions and mutations of the putative binding sites within the *3'reg1-nr2f1a:GFP* reporter vector. Primers used for cloning are listed in **S2 Table**.

  The murine *Foxf1* construct (gift of V. Kalinichenko) was reported previously [75]. The construct contains N-terminal Flag and C-terminal His tags. This construct was subcloned into *pCS2p+DEST1*, a version of *pCS2-Dest1* [97] with a Pst1 site added, using Gateway cloning. The *dnFoxf1* expression construct was generated via fusing the Engrailed transcriptional repressor domain with a 21 bp linker to the 5'-end of the murine *Foxf1* construct using PCR and placed into the *pCS2p+DEST1* vector. Previous work has demonstrated that fusion of the engrailed transcriptional repressor is able to convert transcription factors into dominant negative proteins [98].

## Morpholino (MO), mRNA, and plasmid injections

The *tcf7l1a* (aka *tcf3a*) MO was reported previously [99]. 1 ng of *tcf7l1a* MO was injected. mRNAs for *Foxf1*, *dnFoxf1*, and zebrafish *tcf7l1a* [54] were generated from linearized plasmid using a Sp6 Message Machine kit (Thermo-Fisher, Cat # AM1340). 50 pg of mRNA for *Foxf1* and *dnFoxf1* were injected. 5 pg of zebrafish *tcf7l1a* was injected. All injections were performed at the one-cell stage.

## Generation of transgenic lines

*Tol2* mRNA was generated from linearized plasmid using a Sp6 Message Machine kit (Thermo-Fisher, Cat # AM1340) [96]. For transient transgenesis and the generation of stable transgenic lines, 25 pg of plasmid DNA was co-injected with 25 pg of *Tol2* mRNA. For stable transgenic lines, fish were raised to adulthood and founders were identified via outcrossing to WT fish. The transgenic F1 fish were then selected and raised. At least two independent transgenic lines were initially examined and raised to confirm the equivalency of expression patterns. Only one transgenic line for each stable transgenic reporter was maintained. For experiments, homozygous transgenic *3'reg1:GFP* reporter fish were outcrossed to WT AB/TL fish.

## Drug treatments

All drug treatments were performed on embryos beginning at the 20s stage in 2 mL of embryo water with drugs at the specified concentrations in 3 mL glass vials. 30 embryos/vial were used for all experiments. Treated embryos in vials were placed on a nutator in an incubator at 28.5°C for the duration of the treatments. Concentrations of drugs used for treatments were: 1 μM RA (Sigma R2625), 2.5 μM DEAB (Sigma-Aldrich, Cat # D86256), 5 μM BIO (Tocris, Cat # 3194), 10 μM XAV939 (Tocris, Cat # 3748), 75 μM CYA (Sigma-Aldrich, Cat # C4116), 10 μM SAG (Tocris, Cat # 4366). All the drugs were rinsed out with blue water three times at 48 hpf prior to processing for IHC performed as described below.

## Immunohistochemistry (IHC), cardiomyocyte quantification, and intensity measurements

IHC was performed as has been previously reported [18]. Briefly, embryos at the indicated stages were fixed for 1 hr at RT in 1% formaldehyde/1X PBS in 3 ml glass vials. Embryos were washed 1X in PBS and then 2X in 0.1% saponin/1X PBS, followed by blocking in 0.1% saponin/0.5% sheep serum/1X PBS (Saponin blocking solution) for one hr. Primary antibodies were applied to the embryos in blocking solution overnight at 4°C. Embryos were then gently washed 3X with 0.1% saponin/1X PBS followed by 2 hr incubation with secondary antibodies at RT. Secondary antibodies were washed out with 0.1% saponin/1X PBS. Primary antibodies used were: rabbit anti-Nr2f1a (1:100; custom antibody [12]), rabbit anti-DsRED (1:1000; Clontech, Cat # 632496), chicken anti-GFP (1:250; Invitrogen, Cat # A10262), mouse anti-Amhc (1:10, Developmental Studies Hybridoma Bank, Cat # S46), mouse anti-Mhc (1:10, Developmental Studies Hybridoma Bank, Cat #, MF20). Secondary antibodies used: Goat anti-mouse IgG1 DyLight (1:250, Biolegend, Cat # 409109), Goat anti-chicken IgY Alexa Fluor 488 (1:500, Invitrogen, Cat # A11039), Goat anti-rabbit IgG TRITC (1:100, Southern Biotech, Cat # 4050–03), Goat anti-mouse IgG2b TRITC (Southern Biotech, 1:100, Cat # 1090–03). Cardiomyocytes were quantified as previously reported [46]. Briefly, embryos were mounted ventral side down on coverslips. Images of the hearts were taken using a Nikon A1R Confocal Microscope with a 16X water immersion objective and the resonance scanner. The number of Nr2f1a$^+$ and

*myl7*:DsRed2-NLS$^+$ nuclei in Amhc+ cardiomyocytes were counted in the images using Fiji [100]. The mean intensity of expression for different IHC stainings was measured using ImageJ [101]. For GFP, Nr2f1a, and Amhc staining within the heart a region of interest (ROI) within the images were defined in the individual color panels and the mean gray value of the ROI was measured. The level of staining intensity for GFP within the heart and Nr2f1a within the atria and ventricles in the images was determined by normalizing to mean intensity of Amhc staining in the atria. Embryos were genotyped following imaging and analysis.

### EdU labeling

EdU labeling was carried out using a Click-iT EdU Alexa Fluor Imaging Kit (Molecular Probes, Cat # C10337), as previously described [35]. At 24 hpf, WT, BIO-treated and *Foxf1* mRNA-injected *Tg(myl7:DsRed2-NLS)* embryos were incubated with 10 mM EdU (488) for 30 min on ice. Subsequently, EdU was rinsed and the embryos incubated at 28.5˚C until 48 hpf, to be fixed in 1% formaldehyde in PBS and IHC was performed as described above. Embryos were then post-fixed in 2% formaldehyde in PBS for 1 hr at RT, followed by washes with 1× PBS. The Click-iT reaction was carried out following the manufacturer's guidelines. The hearts were imaged as described above. The number of EdU$^+$/*myl7*:DsRed2-NLS$^+$ cells in each chamber was counted and normalized to the total number of DsRed2-NLS$^+$ cells to show the proliferation index.

### In situ hybridization (ISH)

Whole-mount ISH was conducted similar to what has been previously reported [18]. Briefly, embryos were fixed overnight at 4˚C in 4% paraformaldehyde/1X PBS in 3 ml glass vials. Subsequently, embryos were rinsed with PBS/0.1% Tween-20 (PBST) 3X and then dehydrated with increased percentage of methanol in PBST in series to 100% methanol and stored overnight at 4˚C. Embryos were then rehydrated using a PBST/methanol series to 100% PBST, and then washed 4X in PBST. Digoxygenin-labeled *GFP* (ZDB-EFG-070117-2) riboprobe was used. Anti-Digoxygenin-AP, Fab fragment (Millipore-Sigma, Cat # 11093274910) was used to detect the Digoxygenin-labeled riboprobe in embryos. 3.5μL/mL NBT (Sigma-Roche, Cat # 11383213001) plus 3.5μL/mL BCIP (Sigma-Roche, Cat # 11383221001) in the Staining buffer (0.1M Tris-HCl, pH 9.5/0.1M MgCl2/1M NaCl/0.1% Tween-20) were used for the coloration reaction. Embryos were then rinsed with PBST 3X and dehydrated with a methanol/PBST series to 100% methanol, stored overnight at 4˚C, followed by rehydration using a PBST/methanol series to 100% PBST. The embryos were then put in a glycerol/PBST series to 80% glycerol/PBST for imaging with a Zeiss_v12 M2Bio Stereo Microscope.

### Heart isolations

Hearts from WT and injected zebrafish at 24 hpf were isolated similar to what was previously described [102]. ~500 *Tg(-5.1myl7:EGFP)$^{twu26}$* [93] embryos were anesthetized and transferred to a 1.5 mL microcentrifuge tube. They were then washed on ice 4 times in cold embryo disruption media (EDM, Gibco Leibovitz's L-15 Medium, Cat # 11415114) with 10% fetal bovine serum (FBS, Gibco, Cat # 26140079). Subsequently, embryos were placed in 1.25 mL EDM and triturated for 1 minute using a syringe (1 sec rate) positioned on a ring stand. The fragments were applied to a 105 mM mesh with additional EDM, and immediately to a 40 mM mesh with additional EDM. Finally, the GFP$^+$ hearts were selected utilizing a Zeiss_v12 M2Bio fluorescent stereo microscope and placed in clean EDM. The hearts were pelleted at 1300 rpm per 5 minutes. RNA was extracted using a Single Cell RNA Purification Kit (Norgen Biotek,

Cat # 51800). A SeqPlex RNA Amplification Kit (Sigma-Aldrich, Cat # SEQR) was used to generate cDNA and amplify the library.

### Real Time-quantitative PCR (RT-qPCR)

RT-qPCR experiments were performed as previously described [103]. RT-qPCR for *nr2f1a*, *dkk1a*, *axin1* and *axin2* was carried out using Power SYBR Green PCR Master Mix (Applied Biosystems, Cat # 4368706) in a BioRad CFX-96 PCR machine. Expression levels were standardized to *β-actin* expression [103] and data were analyzed using the Livak $2^{-\Delta\Delta CT}$ Method [104]. Each experiment was performed in triplicate. Primers used are listed in **S3 Table**.

### Generation of zebrafish mutants and crispants

The *3'reg1* deletion mutants and *foxf1* crispants were made using CRISPR/Cas12a (Cfp1), similar to what has been reported [61]. *Acidaminococcus sp. BV3L6 (As)* crRNAs were generated to the 5'- and 3'- sequences of the *3'reg1-nr2f1a* enhancer and the coding region of *foxf1* exon1 using CHOPCHOP (http://chopchop.cbu.uib.no) [62]. The target-specific sequences at the 3'-end of the *As* crRNAs were fused with the common portion of the *As* crRNA with a T7 site at the 5'-end using PCR. The *As* crRNAs were then generated using T7 MEGAScript Kit (Thermo Fisher, Cat # AM1333). A solution of 2 *As* crRNAs (380 pg/μl) were incubated at RT with *As* Cas12a protein (2 μg/μl; IDT, Cat # 10001272) for 20 minutes prior to injection. 760 ng of the crRNA mixture was co-injected along with 4 ng of AsCas12a protein. Multiple pools of 10 embryos were screened for efficacy of the crRNA pairs in generating deletions with PCR. For the *3'reg1* enhancer, sibling embryos from injections that showed efficiency of the crRNAs in creating the appropriate deletions were raised to adulthood. Multiple pools of F1 progeny from founders were then screened for deletions using PCR, followed by cloning and sequencing. A *3'reg1* allele with a 354 bp deletion was identified and subsequently recovered. For the *foxf1* crispants, the *3'reg1*:*GFP* reporter embryos were scored for the presence of GFP and the individual embryos were screened for the efficacy of the deletions following imaging. Oligos used for generation of the sgRNA are listed in **S4 Table**. Primers used for genotyping and testing gRNA efficiency are listed in **S5 Table**.

### Statistical analysis

To determine if two proportions were statistically distinct, we performed Fisher's exact test. To determine if proportions involving more than 2 conditions were statistically different, we performed a Chi-square test. To determine if the qPCR results were significant, we performed a Student's t-test with Welch's correction. To compare 3 or more distinct conditions (cell quantification), we performed an ordinary one-way ANOVA with multiple comparisons. Statistical analyses were performed utilizing GraphPad Prism. A p value $<0.05$ was considered statistically significant for all analysis.

### Supporting information

**S1 Fig. The proximal promoter region of *nr2f1a* promotes broad expression in zebrafish embryos. A)** Schematic of open chromatin in ACs from ATAC-seq showing the putative promoter fragments relative to the *nr2f1a* locus that were analyzed with stable transgenic lines. Image of tracks generated in UCSC genome browser (https://genome.ucsc.edu). **B-F)** Schematics of the *nr2f1a* promoter transgenic constructs and images of representative stable transgenic lines. All the promoter constructs showed broad expression throughout the embryos.

Lateral view with anterior left and dorsal upward. Scale bar: 500 μm.
(TIF)

**S2 Fig. *Nr2f1a* enhancers promote expression in neural tissue and the heart. A)** Schematic of ATAC-seq data in ACs showing the localization of additional putative enhancers that were examined with reporters relative to the *nr2f1a* locus. **B)** VISTA plot showing conservation of the zebrafish *5'reg1-nr2f1a* enhancers with regions in *Callorhinchus milli* (Australian ghostshark), *Xenopus tropicalis* (Tropical clawed frog), *Anolis carolinensis* (Green Anole), *Gallus gallus* (chicken), *Mus musculus* (House mouse), and *Homo sapiens* (human). **C)** Image of *5'reg1*:GFP embryo with expression in the anterior hindbrain and branchial arches at 48 hpf. Lateral view is anterior to the left and dorsal upward. (n = 25). Scale bar: 200 μm. **D)** VISTA plot showing conservation of the zebrafish *3'reg2-nr2f1a* enhancer with regions in *Callorhinchus milli* (Australian ghostshark), *Xenopus tropicalis* (Tropicalis clawed frog), *Anolis carolinensis* (green Anole), *Gallus gallus* (chicken), *Mus musculus* (House mouse), and *Homo sapiens* (human). **E)** Confocal image of heart at 48 hpf from *3'reg2*:GFP embryo stained for *3'reg2*:GFP (green), Vmhc (red), and Amhc (blue). V indicates ventricle. A indicates atrium. (n = 21). Scale bar: 50 μm. **F)** VISTA plot showing conservation of the zebrafish *3'reg3-nr2f1a* enhancer with regions in *Mus musculus* (House mouse) and *Homo sapiens* (human). **G)** Confocal images of heart at 48 hpf from *3'reg3*:GFP embryos stained for *3'reg3*:GFP (green), Vmhc (red), and Amhc (blue). V indicates ventricle. A indicates atrium. Scale bar: 50 μm. **H)** VISTA plot showing conservation of the zebrafish *3'reg4-nr2f1a* enhancer with regions in *Callorhinchus milli* (Australian ghostshark), *Xenopus tropicalis* (Tropical clawed frog), *Gallus gallus* (chicken), *Mus musculus* (House mouse), and *Homo sapiens* (human). **I)** Image of the *3'reg4*:GFP embryo with expression in the medial eye, the anterior brain, the nasal pits, and the first branchial arch at 48 hpf (n = 23). Scale bar: 200 μm. Pink in VISTA plots indicates >50% conservation of regulatory regions with the zebrafish enhancer sequence. Median lines in individual VISTA plots indicate 75% conservation. n indicates the number of embryos examined from a representative clutch. All images are of stable transgenic embryos.
(TIF)

**S3 Fig. Conservation of *nr2f1a* enhancer sequences in other vertebrate species. A-D)** Clustal alignments of *5'reg1-nr2f1a/Nr2f1*, *3'reg2-nr2f1a/Nr2f1*, *3'reg3-nr2f1a/Nr2f1*, *3'reg4-nr2f1a/Nr2f1* between zebrafish and additional vertebrate species: *Callorhinchus milli* (Australian ghostshark), *Xenopus tropicalis* (Tropical clawed frog), *Gallus gallus* (chicken), *Anolis carolinensis* (Green Anole), *Mus musculus* (House mouse), and *Homo sapiens* (human). Turquoise indicates completely conserved nucleotides. Red indicates partially conserved nucleotides.
(TIF)

**S4 Fig. Overlap of *3'reg1*:GFP and Nr2f1a expression within the atrium. A-C″″)** Merged and individual channels of the confocal images of hearts from Fig 1F–1H. Hearts from transgenic *3'reg1*:GFP embryos stained for (**A′-C′**) *3'reg1*:GFP (green), (**A″-C″**) Nr2f1a (magenta), (**A‴-C‴**) Amhc (ACs–blue), and (**A″″-C″″**) Mhc (pan-cardiac–red). 36 hpf (n = 4), 48 hpf (n = 4), 72 hpf (n = 4). v–ventricle. a–atrium. *3'reg1*:GFP in the atria of the hearts (white arrowheads). Images are frontal views with the arterial pole up. n indicates the number of embryos examined for representative experiment. Scale bar: 50 μm.
(TIF)

**S5 Fig. The NHR site within *3'reg1* is not required to promote expression or responsive to loss of RA signaling. A)** Clustal alignment of *3'reg1* with the putative NHR site highlighted, as well as Foxf1 and Lef/Tcf sites as shown in Fig 1. **B)** Schematics of *3'reg1*:GFP reporter constructs. Foxf sites (blue), Lef/Tcf site (purple), NHR site (orange), deleted NHR site (black). **C)**

Sequences showing WT NHR site and deletion of the NHR site in the *3'reg1:GFP* constructs. Deletion of the NHR site did not affect expression within the heart relative to the WT *3'reg1*: *GFP* construct (n = 48) nor did treatment with the RA signaling inhibitor DEAB (N = 33). **D)** The percentage of transient transgenic embryos with reporter expression in the atria of their hearts. *3'reg1:GFP* (n = 21); *3'reg1-ΔNHR:GFP* (n = 26). **E)** The percentage of stable *3'reg1:GFP* embryos with no, atrial, and pan-cardiac reporter expression in their hearts follow treatment with RA or the RA signaling inhibitor DEAB. Control (n = 19); RA (n = 25); DEAB (n = 26). (TIF)

**S6 Fig. Modulation of Wnt signaling affects the intensity of Nr2f1a expression within the heart. A-C""")** Merged and individual channels of the confocal images of hearts from Fig 4F–4H. Hearts from transgenic *3'reg1:GFP* embryos stained for (**A'-C'**) Nr2f1*a* (green), (**A"-C"**) Mhc (pan-cardiac–red) (magenta), and (**A'''-C'''**) Amhc (ACs–blue). VCs with a low level of Nr2f1a within the nuclei (white arrowheads). Images are frontal views with the arterial pole up. Scale bar: 50 μm. **D)** Normalized intensity of Nr2f1a staining in atria of hearts from control, BIO-, and XAV-treated embryos. Control (n = 7); BIO (n = 10); XAV (n = 8). **E)** Normalized intensity of Nr2f1a staining in ventricles of hearts from control and BIO treated embryos. Control (n = 8); BIO (n = 7). Error bars in graphs indicate s.e.m. * indicate $P < 0.05$, **** indicate $P < 0.0001$. (TIF)

**S7 Fig. Wnt signaling and Foxf1 promote increased *nr2f1a* expression in hearts by 24 hpf. A,B)** RT-qPCR for *nr2f1a* on cDNA from isolated hearts at 24 hpf of embryos treated with BIO and XAV, and injected with *Foxf1* mRNA and *dnFoxf1* mRNA. Fold difference is relative to *β-actin*. Control in A,B is the same. Error bars in graphs indicate s.e.m. * indicate $P < 0.03$, **** indicate $P < 0.0001$. (TIF)

**S8 Fig. Manipulation of Wnt signaling and Foxf1 does not impact VC number. A)** The number of VCs (*myl7*:DsRed2-NLS+/Amhc- cardiomyocytes) within the hearts of control, BIO-, and XAV-treated embryos. Control (n = 9); BIO (n = 10); XAV (n = 8). **B)** The number of VCs (*myl7*:DsRed2-NLS+/Amhc- cardiomyocytes) within the hearts of control, *Foxf1* mRNA, and *dnFoxf1* mRNA injected embryos. Control (n = 10); *Foxf1* mRNA (n = 13); *dnFoxf1* mRNA (n = 11). Error bars in graphs indicate s.e.m. (TIF)

**S9 Fig. Manipulation of Wnt signaling and Foxf1 does not impact cardiomyocyte proliferation. A-C)** Confocal images of hearts from control, BIO-treated, and *Foxf1* mRNA injected *myl7:DsRed2-NLS* embryos at 48 hpf that were pulsed with EdU at 24 hpf. EdU (green), *myl7*: DsRed2-NLS (red), Amhc (blue). Arrows indicate co-labeled EdU+ and DsRed2-NLS+ cardiomyocyte nuclei. Scale bar: 50 μm. **D)** Proliferation Index of atrial and ventricular cardiomyocytes from Control, BIO treated, and *Foxf1* mRNA injected *myl7:DsRed2-NLS* embryos at 48 hpf that were pulsed with EdU at 24 hpf. Ventricle—Control (n = 5), BIO (n = 6), *Foxf1* mRNA (n = 5). Atrium–Control (5), BIO (n = 6), *Foxf1* mRNA (n = 5). ns indicates not significance difference relative to controls. Error bars in graphs indicate s.e.m. (TIF)

**S10 Fig. Concurrent manipulation of Tcf7l1a and Wnt signaling in *3'reg1:GFP* reporter embryos. A-I)** Confocal images of hearts at 48 hpf from *3'reg1:GFP* control, *tcf7l1a* mRNA injected, *tcf7l1a* MO injected, BIO-treated, BIO-treated and *tcf7l1a* mRNA injected, BIO-treated and *tcf7l1a* MO injected, XAV-treated, XAV-treated and *tcf7l1a* mRNA injected,

XAV-treated and *tcf7l1a* MO injected embryos. *3'reg1:GFP* embryos stained for *3'reg1*:GFP (green), Vmhc (red), and Amhc (blue). **J)** The percentage of *3'reg1*:GFP embryos with atrial, pan-cardiac, and lacking expression in their hearts. *3'reg1:GFP* control (n = 17), *tcf7l1a* mRNA injected (n = 20), *tcf7l1a* MO injected (n = 20), BIO-treated (n = 26), BIO-treated and *tcf7l1a* mRNA injected (n = 32), BIO-treated and *tcf7l1a* MO injected (n = 34), XAV-treated (n = 24), XAV-treated and *tcf7l1a* mRNA injected (n = 24), XAV-treated and *tcf7l1a* MO injected embryos (n = 30). \*\*\*\* indicate P < 0.0001.
(TIF)

**S11 Fig. Concurrent manipulation of Tcf7l1a and Wnt signaling in *myl7:DsRed2-NLS* embryos. A-I)** Confocal images of hearts at 48 hpf from *myl7:DsRed2-NLS* control, *tcf7l1a* mRNA injected, *tcf7l1a* MO injected, BIO-treated, BIO-treated and *tcf7l1a* mRNA injected, BIO-treated and *tcf7l1a* MO injected, XAV-treated, XAV-treated and *tcf7l1a* mRNA injected, XAV-treated and *tcf7l1a* MO injected embryos. *3'reg1:GFP* embryos stained for *myl7*: *DsRed2-NLS* (red) and Amhc (blue). **J)** The number of *myl7*:DsRed2-NLS⁺/Amhc⁺ cardio-myocytes (ACs) within the hearts of Control (n = 8), *tcf7l1a* mRNA injected (n = 7), *tcf7l1a* MO injected (n = 7), BIO-treated (n = 8), BIO-treated and *tcf7l1a* mRNA injected (n = 7), BIO-treated and *tcf7l1a* MO injected (n = 8), XAV-treated (n = 9), XAV-treated and *tcf7l1a* mRNA injected (n = 7), XAV-treated and *tcf7l1a* MO injected *myl7:DsRed2-NLS* embryos (n = 6). \*\*\*\* indicate P < 0.0001.
(TIF)

**S12 Fig. Loss of *foxf1* in zebrafish leads to a reduction in *3'reg1:GFP* expression within hearts. A)** Schematic showing the location of the guides (arrow) spaced ~200 bp apart in the first exon of zebrafish *foxf1*. **B)** PCR showing the efficacy of guides in creating an ~200 bp deletion and eliminating the WT band for *foxf1* within representative injected *3'reg1:GFP* embryos. **C)** Confocal images of hearts from control and *foxf1* CRISPR-Cas12 injected transgenic *3'reg1*: *GFP* embryos stained for *3'reg1*:GFP (green), Vmhc (red), and Amhc (blue). Scale bars: 50 μm. **D)** The percentage of control uninjected and *foxf1* crispant *3'reg1:GFP* embryos with expression in the heart. Control (n = 44); *Foxf1* crispant (n = 48). \*\*\*\* indicate P < 0.0001.
(TIF)

**S13 Fig. Modulation of Foxf1 affects the intensity of Nr2f1a expression within the heart. A-C'''')** Merged and individual channels of the confocal images of hearts from Fig 6F–6H. Hearts from transgenic *3'reg1:GFP* embryos stained for (**A'-C'**) Nr2f1*a* (green), (**A''-C''**) Mhc (pan-cardiac–red) (magenta), and (**A'''-C'''**) Amhc (ACs–blue). Scale bar: 50 μm. **D)** Normalized intensity of Nr2f1a staining in atria of hearts from control, *Foxf1* mRNA, and *dnFoxf1* mRNA injected embryos. Control (n = 5); *Foxf1* mRNA (n = 7); *dnFoxf1* mRNA (n = 8). **E)** Normalized intensity of Nr2f1a staining in ventricles of hearts from control and *Foxf1* mRNA injected embryos. Control (n = 6); *Foxf1* mRNA-injected (n = 6). Error bars in graphs indicate s.e.m. \* indicate P < 0.05, \*\*\*\* indicate P < 0.0001.
(TIF)

**S14 Fig. Hh signaling modestly represses *3'reg1:GFP* expression within the hearts. A-G)** Confocal images of hearts from untreated control, CYA-treated, SAG-treated, XAV-treated, BIO-treated, XAV+CYA-treated, and BIO+SAG-treated transgenic *3'reg1:GFP* embryos stained for *3'reg1*:GFP (green), Vmhc (red), and Amhc (blue). **H)** The percentage of control, CYA-treated, SAG-treated, XAV-treated, BIO-treated, XAV+CYA-treated, and BIO+SAG-treated *3'reg1:GFP* embryos that had expression in the atria, inhibited expression, expanded expression within the atria, and pan-cardiac expression. Control (n = 31); CYA (n = 44); SAG (n = 98). XAV-treated (n = 48), BIO-treated (n = 28), XAV+CYA-treated (n = 99), and BIO

+SAG-treated (n = 109). **I,J)** Confocal images of hearts from WT sibling and *smo* mutant *3'reg1*:*GFP* embryos stained for *3'reg1*:GFP (green), Vmhc (red), and Amhc (blue). **K)** The percentage of WT and *smo 3'reg1*:*GFP* embryos that had expanded reporter expression within their atria. Control (n = 24); *smo* (n = 51). Scale bars: 50 μm. Error bars in graph indicate s.e. m. **** indicate P < 0.0001. ns indicates not a statistically significant difference.
(TIF)

**S15 Fig. Hh signaling does not affect Nr2f1a expression within the hearts. A-G)** Confocal images of hearts from untreated control, CYA-treated, SAG-treated, XAV-treated, BIO-treated, XAV+CYA-treated, and BIO+SAG-treated embryos stained for Nr2f1a (green), Mhc (red), and Amhc (blue). Scale bar: 50 μm. **H)** The number of Nr2f1a⁺/Amhc⁺ cardiomyocytes in the hearts of untreated control, CYA-treated, SAG-treated, XAV-treated, BIO-treated, XAV +CYA-treated, and BIO+SAG-treated embryos. Control (n = 5), CYA (n = 8), SAG (n = 7), XAV-treated (n = 7), BIO-treated (n = 10), XAV+CYA-treated (n = 6), and BIO+SAG-treated (n = 7). Error bars in graph indicate s.e.m. **** indicate P < 0.0001. ns indicates not a statistically significant difference.
(TIF)

**S16 Fig. Foxf1 sites are required for *3'reg1* reporter expression. A)** Schematics of *3'reg1*:*GFP* reporter constructs with WT, mutated Foxf-A+mutated Tcf sites, mutated Foxf-B+mutated Tcf sites, mutated Foxf-C+mutated Tcf sites, and mutated of all 3 Fox sites (Foxf-A-B-C) + mutated Tcf sites. WT Foxf sites (blue). WT Tcf site (purple). Mutated sites (red). Mutations were made as in Figs 2 and 5. **B)** The percentage of transient transgenic embryos with reporter expression in the atria. *3'reg1* (n = 34); *mFoxf-A+Tcf* (n = 80); *mFoxf-B+Tcf* (n = 50); *mFoxf-C +Tcf* (n = 45); *mFoxf-A-B-C+Tcf* (n = 28). **** indicate P < 0.0001.
(TIF)

**S17 Fig. Foxf1 promotes increased Wnt signaling in hearts. A,B)** RT-qPCR for *axin1*, *axin2*, and *dkk1a* in isolated hearts from 24 hpf embryos injected with *Foxf1* mRNA and *dnFoxf1* mRNA. Fold difference is relative to *β-actin*. Error bars in graphs indicate s.e.m. ** indicates P < 0.002.*** indicates P < 0.001, **** indicates P < 0.0001.
(TIF)

**S18 Fig. Foxf1 promotes expression of a Wnt signaling reporter. A-C""')** Merged and individual channels of the confocal images of hearts from control, *Foxf1* mRNA injected, and *dnFoxf1* mRNA injected *7xTcf.Xla-Sia*:*GFP* embryos at 34 hpf. **(A'-C')** *7xTcf.Xla-Sia*:*GFP* (green), **(A"-C")** Mhc (red), and **(A"'-C"')** Amhc (blue). Scale bar: 50 μm. **D)** Percentage of Control, *Foxf1* mRNA injected, and *dnFoxf1* mRNA injected *7xTcf.Xla-Sia*:*GFP* embryos at 34 hpf with normal, increased, or decreased expression near and in the hearts. Control (n = 35), *Foxf1* mRNA (n = 97), *dnFoxf1* mRNA (n = 106).
(TIF)

**S19 Fig. Zebrafish *3'reg1*⁻/⁻ embryos. A)** PCR from WT and *3'reg1* embryos. The WT *3'reg1* PCR product is 594 bp. The PCR product for the *3'reg1* deletion is 354 bp. **B)** *3'reg1* sequence showing the deleted sequence (blue). **C,D)** Representative WT and *3'reg1*⁻/⁻ embryos at 72 hpf. Lateral views with anterior leftward and dorsal upward. *3'reg1*⁻/⁻ embryos do not have an overt phenotype. Scale bar: 500 μm.
(TIF)

**S20 Fig. Intensity of Nr2f1a expression is reduced in atria of *3'reg1*⁻/⁻ embryos. A-B"')** Merged and individual channels of the confocal images of hearts from *3'reg1*⁻/⁻ embryos. **(A', B')** Nr2f1a (green), **(A",B")** Mhc (red), **(A"',B"')**, and Amhc (blue). Scale bar: 50 μm. **C)**

Normalized intensity of Nr2f1a staining in atria of hearts from WT sibling and *3'reg1*$^{-/-}$ embryos. WT sibling (n = 12); *3'reg1*$^{-/-}$ (n = 8). Error bars in graphs indicate s.e.m. **** indicate P < 0.0001.
(TIF)

**S21 Fig.** *3'reg1-nr2f2* **enhancer is conserved in gnathostomes. A)** Schematic of ATAC-seq data in ACs showing the localization of the putative *3'reg1-nr2f2* enhancer relative to the *nr2f2* locus. Image of tracks generated in UCSC genome browser (https://genome.ucsc.edu). **B)** Clustal alignment of *3'reg1-nr2f2* from *Danio rerio* (zebrafish), *Callorhinchus milli* (Australian ghostshark), *Xenopus tropicalis* (Tropical clawed frog), *Anolis carolinensis* (Green Anole), *Gallus gallus* (chicken), *Mus musculus* (House mouse), and *Homo sapiens* (human). Turquoise indicates completely conserved nucleotides. Red indicates partially conserved nucleotides. **C)** VISTA plot showing conservation of the zebrafish *3'reg1-nr2f2* enhancer with regions in *Callorhinchus milli* (Australian ghostshark), *Xenopus tropicalis* (Tropicalis clawed frog), *Anolis carolinensis* (green Anole), *Gallus gallus* (chicken), *Mus musculus* (House mouse), and *Homo sapiens* (human). Pink indicates >50% conservation of regulatory regions with zebrafish *3'reg1*. Median lines in individual VISTA plots indicate 75% conservation.
(TIF)

**S1 Table. Zebrafish** *nr2f1a* **promoter and 5'UTR sequences analyzed in transgenic embryos with GFP reporters.**
(DOCX)

**S2 Table. Primers used for cloning enhancers, generating deletions and mutations, and generating expression constructs.**
(XLSX)

**S3 Table. Primers used for RT-qPCR.**
(XLSX)

**S4 Table. Sequences of the oligos used to generate the CRISPR-Cas12AS sgRNAs.**
(XLSX)

**S5 Table. Primers used for genotyping.**
(XLSX)

**S6 Table. Values of data used to generate graphs within the figures.**
(XLSX)

## Acknowledgments

We thank Kendall Martin for technical assistance, training, and reading of the manuscript.

## Author Contributions

**Conceptualization:** Ugo Coppola, Joshua S. Waxman.

**Data curation:** Ugo Coppola, Bitan Saha, Joshua S. Waxman.

**Formal analysis:** Ugo Coppola, Bitan Saha, Jennifer Kenney.

**Funding acquisition:** Ugo Coppola, Joshua S. Waxman.

**Investigation:** Ugo Coppola, Bitan Saha, Jennifer Kenney.

**Methodology:** Ugo Coppola, Joshua S. Waxman.

**Resources:** Joshua S. Waxman.

**Supervision:** Joshua S. Waxman.

**Validation:** Ugo Coppola, Bitan Saha.

**Visualization:** Ugo Coppola, Joshua S. Waxman.

**Writing – original draft:** Ugo Coppola, Joshua S. Waxman.

**Writing – review & editing:** Ugo Coppola, Joshua S. Waxman.

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
