## [Decision Letter · Decision Letter 0]

19 Apr 2024

Dear Dr Waxman,

Thank you very much for submitting your Research Article entitled 'A Foxf1-Wnt-Nr2f1 cascade promotes atrial cardiomyocyte differentiation in zebrafish' to PLOS Genetics.

The manuscript was fully evaluated at the editorial level and by independent peer reviewers. The reviewers appreciated the attention to an important problem, but raised some substantial concerns about the current manuscript. Based on the reviews, we will not be able to accept this version of the manuscript, but we would be willing to review a much-revised version. It is worth noting that the referees are in general agreement regarding the revisions required. We cannot, of course, promise publication at that time.

If you decide to revise the manuscript for further consideration at PLOS Genetics, please aim to resubmit within the next 60 days, unless it will take extra time to address the concerns of the reviewers, in which case we would appreciate an expected resubmission date by email to plosgenetics@plos.org.

We are sorry that we cannot be more positive about your manuscript at this stage. Please do not hesitate to contact us if you have any concerns or questions.

Yours sincerely,

Frank L Conlon

Academic Editor

PLOS Genetics

Gregory P. Copenhaver

Section Editor

PLOS Genetics

Reviewer's Responses to Questions

**Comments to the Authors:**

Reviewer #1: In the manuscript authored by Coppola et al., the researchers identified a conserved enhancer 3’ to the nr2f1a locus, named 3’reg1-nr2f1a (3’reg1), that was able to drive reporter gene expression within ACs of stable transgenic lines. The authors also identified a Lef/Tcf binding site and multiple Foxf binding sites within the enhancer, which, when mutated, could lead to ectopic or diminished reporter gene expression in the developing heart. Further gain- and loss-of-function experiments manipulating Wnt signaling and Foxf1 activity supported the observations made by mutating the respective DNA binding motifs. The authors suggest that 3’reg1 acts as the enhancer element promoting Nr2f1a expression in ACs; however, embryos with a homozygous deletion of the endogenous 3’reg1 fragment showed no overt defects and did not exhibit affected Nr2f1a expression in the heart. Therefore, the overall statement that “our data support that downstream members of a conserved regulatory network involving Wnt signaling and Foxf1 function on a nr2f1a enhancer to promote AC differentiation in the zebrafish heart” is not sufficiently supported by data presented in the manuscript.

Reviewer #2: In this manuscript, Coppola et al investigated the regulation of Nr2f1a transcription factor in the differentiation of vertebrate atrial cardiomyocytes (ACs). They identified an enhancer, named 3’reg1, located 3’ to the nr2f1a locus, which promotes expression within ACs. This enhancer contains binding sites for Lef/Tcf and Foxf transcription factors. Mutations in these sites affect the enhancer's activity. The authors proposed a model where Wnt signaling, downstream of Foxf1, relieves Tcf7l1a-mediated repression of the enhancer, promoting AC differentiation. Hh signaling, although known for its role in venous SHF differentiation, only modestly affects the enhancer's activity and does not impact AC differentiation. The authors found that Nr2f1a functions downstream of Wnt signaling and Foxf1. CRISPR-mediated deletion of the enhancer abolishes the ability of Wnt signaling and Foxf1 to enhance AC differentiation, suggesting its essential role in the process. Overall, this study outlines a new regulatory network involving Wnt signaling and Foxf1 that promotes AC differentiation through the regulation of zebrafish nr2f1a expression via a conserved enhancer. It's a comprehensive study employing both gain- and loss-of-function assays, with high data quality and clear writing. However, several concerns need addressing.

1. The extent to which the 3’reg1:GFP reporter reflects endogenous nr2f1a expression remains unclear. Are they expressed in the same cells with similar dynamics during heart development? How many stable lines were analyzed?

2. Figure 2F and Figure 4C show that mutating Tcf and Foxf binding sites did not fully abolish atrial-specific GFP expression, suggesting that these binding sites may not be essential for the atrial activity of the enhancer. Please clarify. Generation of stable transgenic lines may be necessary for sorting this out. Also, analysis of double mutants of the Tcf and Foxf binding sites may be informative to dissect any additive and/or redundant functions of these binding sites.

3. The authors claimed that Wnt signaling activates the transgenic 3’reg1:GFP reporter and promote a surplus of differentiated ACs, likely through derepression of Tcf7l1a. However, rescue experiments using tcf7l1a under manipulated wnt conditions are needed.

4. Lines 266-268, “We found that simultaneously inhibiting or activating Wnt and Hh signaling on decreased and increased expression of the 3’reg1:GFP reporter within the heart, similar to inhibiting or activation of Wnt signaling alone (S7L-O Fig.).” Control groups for inhibiting or activating Wnt signaling alone are lacking in the experiments. Comparisons with the results from other experiments may not be accurate.

5. Lines 268-270, “Therefore, our data suggest that in contrast to mammalian venous pole differentiation Hh signaling is likely not affecting AC differentiation at these stages of cardiogenesis nor is it promoting Nr2f expression in ACs upstream of Wnt or Foxf1 in zebrafish embryos.” The experiments referred here in Fig. S7L-O were performed on the 3’reg1:GFP reporter, not endogenous nr2f1a expression, which should be tested to support claims about Hh signaling.

6. Line 177, the negative data should be shown.

7. Line 185, Please correct this sentence “we found that either deletion or targeted mutation of the Lef/Tcf site both resulted in…”.

Reviewer #3: The manuscript by Coppola et al describes identification of an enhancer that regulates nr2f1a expression during zebrafish atrial cardiomyocyte differentiation. Authors demonstrate that the conserved 3’ enhancer restricts GFP reporter expression to atrial cardiomyocytes. They show that the enhancer is regulated by foxf1 and wnt signaling, and demonstrate their roles in restricting the number of atrial cardiomyocytes. Altogether, the work establishes the regulatory network that involves wnt and foxf1 signaling that function through nr2f1a to promote differentiation of atrial cardiomyocytes in zebrafish.

Overall the results are novel and significant, and they provide mechanistic insight into pathways involved in cardiac differentiation. These findings will be of high interest to many researchers working on cardiac specification and regeneration. The experiments are well performed and documented. Major weaknesses are the lack of quantification of expression intensities in different treatments and the fact that that different experimental manipulations and treatments appear to have different effects on the reg1:GFP enhancer and endogenous nr2f1a expression (see Point #3). It is suggested that the authors address the following points.

1. It would be helpful to show GFP channel separately in Fig. 1F-H. It is not clear if GFP is completely absent from the ventricle, or if it is expressed at a low level.

2. The expansion of reg1:GFP into the ventricle region is very clear in TCF site-deleted and tcf7l1a MO embryos in Fig 2C-I. However, it also appears that the intensity of atrial expression of reg1:GFP may be also increased. It would be helpful to perform some quantification of expression level to test if GFP intensity is higher in mTCF, Delta_TCF and tcf7l1a MO embryos.

3. In many experiments there is a discrepancy between the treatment effect on enhancer reporter expression and the endogenous nr2f1a expression. For example, BIO treatment resulted in ectopic expansion of reg1:GFP expression into the ventricle (Fig. 3B). If this 3’ enhancer is responsible for restricting nr2f1a expression into the ventricle, then a similar effect would be expected on nr2f1a expression. However, a different effect, an increase in the atrial cardiomyocyte numbers, is observed, based on nr2f1a immunostaining analysis (Fig. 3F). Similar discrepancies are apparent in many other experiments (Fig. 4, for example). It is unclear why the treatments would result in a different effect on the reporter and endogenous gene expression. It would be helpful for the authors to consider two possible explanations: a) Perform HCR / FISH analysis for endogenous nr2f1a mRNA expression in BIO, XAV or foxf1 mRNA injected embryos. It is possible that nr2f1a mRNA expression could be expanded into the atrium in BIO treated or foxf1 mRNA injected embryos but this is not apparent by immunostaining due to other posttranscriptional mechanisms involved in restricting protein localization. b) Quantify fluorescence intensity / expression level of atrial nr2f1a expression (either protein, or even better, mRNA), and 3’reg1:GFP expression intensity in the atrium. Based on the provided images it appears that the atrial reg1:GFP fluorescence is higher in BIO treated or foxf1 mRNA injected embryos, and lower in XAV or dnFoxf1 mRNA injected embryos. It also appears that Nr2f1a immunostaining intensity may follow also a similar pattern, being higher in BIO or Foxf1 mRNA injected embryos, and lower in XAV treated or dnFoxf1 mRNA injected embryos. If this is true, then Wnt inhibition / activation or foxf1 could affect reg1:GFP and endogenous Nr2f1a expression in a similar way, by regulating its expression level in the atrium. In my opinion, this discrepancy is currently the major limitation of this study.

4. The authors argue that foxf1 is upstream of Wnt signaling, in part, because foxf1 mRNA promoted a modest increase in the expression of axin1 and decrease in dkk1a expression. However, this qPCR analysis was performed using whole embryos, therefore it is unclear if the same effect was observed in the heart or atrium. It would be helpful to examine the effect on Wnt signaling in foxf1 overexpressing and / or foxf1 inhibited embryos using Wnt reporter line. Alternatively, HCR / FISH approaches could be used to stain and quantify the level of wnt reporter genes in the heart.

5. It would be helpful to quantify atrial nr2f1a expression level (HCR / FISH or immunostaining) in 3’reg1 enhancer deleted embryos. nr2f1a expression level could be altered without affecting the number of atrial cardiomyocytes.

Reviewer #4: The study by Coppola et. al. is focused characterizing the regulatory elements that drive expression of the nr2f1a, a transcription factor that is important for heart development. This study focuses on the cis-regulatory elements (CRE) that induce expression in atrial cardiomyocytes (AC) in zebrafish. The authors initially determined that the nr2f1a promoter is not sufficient to regulate GFP reporter expression in the atria. This led them to compare ATAC-seq data from several species to identity open chromatin domains. At least 3 open regions were identified at the 3’ end of nr2f1a gene locus and they carefully determined that one specific region, 3’reg1, was sufficient to regulate GFP expression in AC. The 3’reg1 CRE contains conserved binding sites for tcf/lef1, foxf1 and nuclear hormone receptor (NHR) transcription factors. The authors revealed that both Wnt signaling and a Forkhead transcription factor, Foxf1 play a role in regulating nr2f1a expression through the 3’reg1 CRE, while the NHR domain was not required. The authors determine that the Wnt pathway functions downstream or at least in parallel with foxf1 in regulating nr2f1a expression in ACs closest to the venous pole.

To support these findings in functional studies, the authors showed that ectopic expression of Foxf1 from mouse or activation of Wnt signaling was sufficient to expand the AC population without disturbing the ventricle cardiomyocyte population (VC). These observations were supported by blocking foxf1 activity with dominant negative foxf1 and blocking Wnt activity resulted in decreased AC populations.

The authors also addressed whether the absence of foxf1 is required for regulating nr2f1a expression in the heart via the 3’reg1. Here loss-of-foxf1 in F0 embryos showed that GFP fluorescence is decreased. To formally show that nr2f1a expression in the atrial requires 3’reg1, the authors deleted this CRE. In this situation, 3’reg1 deletion did not result in heart defects, but activation of Wnt or injection of Foxf1 mRNA did not expand AC populations. The authors conclude that the 3’reg1 CRE is evolutionary conserved and serves as an important regulator of nr2f1a expression in the atria that is dependent on Wnt and foxf1 activity.

The authors could solidify these studies with these suggestions.

- Correlating nr2f1a protein expression and transgene reporter is not characterized. There seems to be a discrepancy of nr2f1a protein expression and ectopic GFP expression, particularly in the ventricle. For example, in Figure 3 F, & G there appears to be expression of Nr2f1 protein throughout the atrial, but it is not stated if Nr2f1 is induced in the ventricle as this was not scored. The authors should determine if treatment with BIO not only increase Nr2f1a positive cells in the atrial but are there ectopic expression of the protein in the ventricle.

- This could also be extended to the injection of Foxf1 mRNA in Figure 4I. Although the number of AC increased, there was no decrease in VCs. Is it known if there is increased proliferation of only AC cells in these injected conditions? Or is there expansion of cardiogenic domains earlier and this only expanded atrial progenitors? The authors should look earlier for expansion of ACs and determine if there is an increase in proliferation in the atria.

- Knockdown of tcf7l1a expand GFP expression (Figure 2H) in the ventricle but also increase Nr2f1a protein in the ventricle?

- Foxf1 crispants or morpholinos should be used as another assay for loss of foxf1 activity. The use of dominant negative that involves engrailed repressor can sometimes cause a gain of function phenotype that is independent of endogenous foxf1 activity. The authors should also cite original studies that used Engrailed Repressor domain to create dominant negatives.

**Have all data underlying the figures and results presented in the manuscript been provided?**

Reviewer #1: Yes

Reviewer #2: Yes

Reviewer #3: Yes

Reviewer #4: Yes

PLOS authors have the option to publish the peer review history of their article (what does this mean?). If published, this will include your full peer review and any attached files.

Reviewer #1: No

Reviewer #2: No

Reviewer #3: No

Reviewer #4: No

---

## [Decision Letter · Decision Letter 1]

21 Oct 2024

Dear Dr Waxman,

We are pleased to inform you that your manuscript entitled "A Foxf1-Wnt-Nr2f1 cascade promotes atrial cardiomyocyte differentiation in zebrafish" has been editorially accepted for publication in PLOS Genetics. Congratulations!

Yours sincerely,

Frank L Conlon

Academic Editor

PLOS Genetics

Gregory P. Copenhaver

Section Editor

PLOS Genetics

Comments from the reviewers (if applicable):

Reviewer's Responses to Questions

**Comments to the Authors:**

Reviewer #1: The authors chose not to address my comments, and under these conditions, I don't have any further input to provide.

Reviewer #2: The authors have addressed my major concerns, and I have no further comments.

Reviewer #3: Authors have appropriately addressed my previous comments during the revision, and the manuscript has been greatly improved. I do not have any further concerns.

Reviewer #4: In the revised submission, Coppola et al. have included new experiments that address my comments. Here they included new experiments to show that proliferation was not altered following Foxf1 mRNA injection or with BIO treatment. In addition, the authors address the role of Tcf7l1a-depeleted embryos on the regulation of Nr2f1a+ cardiomyocytes. These experiments address the suggestions I made.

Overall this is a detailed study on the regulation of atrial cardiomyocyte differentiation as indicated by Nr2f1a expression in the context of Wnt signaling and the forkhead transcription factor, Foxf1.

**Have all data underlying the figures and results presented in the manuscript been provided?**

Reviewer #1: Yes

Reviewer #2: Yes

Reviewer #3: Yes

Reviewer #4: Yes

PLOS authors have the option to publish the peer review history of their article (what does this mean?). If published, this will include your full peer review and any attached files.

Reviewer #1: No

Reviewer #2: No

Reviewer #3: No

Reviewer #4: No

**Data Deposition**

http://datadryad.org/submit?journalID=pgenetics&manu=PGENETICS-D-24-00293R1

**Press Queries**

---

## [Editor Report · Acceptance letter]

29 Oct 2024

PGENETICS-D-24-00293R1 

A Foxf1-Wnt-Nr2f1 cascade promotes atrial cardiomyocyte differentiation in zebrafish 

Dear Dr Waxman, 

We are pleased to inform you that your manuscript entitled "A Foxf1-Wnt-Nr2f1 cascade promotes atrial cardiomyocyte differentiation in zebrafish" has been formally accepted for publication in PLOS Genetics! Your manuscript is now with our production department and you will be notified of the publication date in due course.

With kind regards,

Anita Estes

PLOS Genetics

On behalf of:
